# TimeWalker: Personalized Neural Space for Lifelong Head Avatars

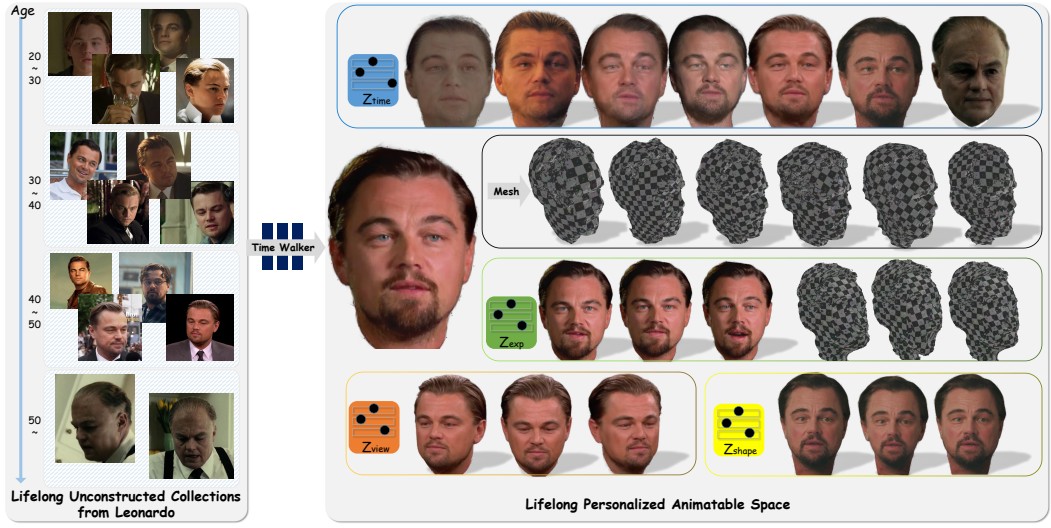

Figure 1: **TimeWalker.** Given a set of unstructured data from the Internet or photo collection across years, we build a personalized neural parametric morphable model, *TimeWalker*, towards replicating a life-long 3D head avatar of a person. With the TimeWalker, we can control and animate one's avatar in terms of shape, expression, viewpoint, and appearance across his/her different age periods. In this Figure, We show Leonardo Dicaprio's life-long avatar reconstructed and animated by our proposed model. More results are presented on the project page: https://timewalker2024.github.io/timewalker.github.io/

## Abstract

We present **TimeWalker**, a novel framework that models realistic, full-scale 3D head avatars of a person on *lifelong scale*. Unlike current human head avatar pipelines that capture a person's identity only at the momentary level (*i.e.,* instant photography, or short videos), TimeWalker constructs a person's comprehensive identity from unstructured data collection over his/her various life stages, offering a paradigm to achieve full reconstruction and animation of that person at different moments of life. At the heart of TimeWalker's success is a novel neural parametric model that learns personalized representation with the disentanglement of shape, expression, and appearance across ages. Central to our methodology are the concepts of two aspects: (1) We track back to the principle of modeling a person's identity in an additive combination of his/her average head representation in the canonical space, and moment-specific head attribute representations driven from a set of neural head basis. To learn the set of head basis that could represent the comprehensive head variations of the target person in a compact manner, we propose a **Dy**namic **N**eural **Ba**sis-Blending **Mo**dule (**Dynamo**). It dynamically adjusts the number and blend weights of neural head bases, according to both shared and specific traits of the target person over ages. (2) We introduce **Dyna**mic **2**D Gaussian **S**platting (**DNA-2DGS**), an extension of Gaussian splatting representation, to model head motion deformations like facial expressions without losing the realism of rendering and reconstruction of full head. DNA-2DGS includes a set of controllable 2D oriented planar Gaussian disks that utilize the priors from a parametric morphable face model, and move/rotate with the change of expression. Through extensive experimental evaluations, we show TimeWalker's ability to reconstruct and animate avatars across decoupled dimensions with realistic rendering effects, demonstrating a way to achieve personalized "time traveling" in a breeze.

## 1  INTRODUCTION

*"As we grow older, our life stories become our identity. This ongoing narrative of the self is constructed from the past and anticipated future."* – James E. Birren, Psychologist, 1996.

What forms a person's identity? In the realm of Computer Graphics and Vision, researchers have traditionally considered an individual's shape as invariant, assuming it to represent the person's identity. This assumption has driven significant advancements in human faces and head modeling over the decades. From classic 3DMMs (3D morphable models) Blanz & Vetter (1999) and its follow-up line of work Li et al. (2017); Paysan et al. (2009), to current advanced neural representations for head modeling Yenamandra et al. (2021); Giebenhain et al. (2023); Qian et al. (2023); Hong et al. (2022), 3D head avatars become increasingly lifelike, serving as *momentary* replicas of humans.

However, sociology and psychology provide a different perspective to answer the question of identity formation – the fields emphasis the idea that a person's identity is shaped by a continuous process influenced by various lifestages and experiences of self over time, rather than a single moment in time. Since the $1960s$, sociologists and psychologists (*e.g.,*Waterman (1982); Kroger (2007)) have recognized the intrinsic value of *lifelong* construction of self-identity, and have developed several cornerstone theories upon this basis.

Motivated by the critical identity formation gap, in this work, we aim to explore a paradigm of *modeling 3D head avatars on a person's lifelong scale*. This new problem definition introduces three major challenges to head avatar modeling: (1) **Long-horizon Identity Consistency Preserving.** The life-long modeling breaks the long-lasting assumption of the shape invariant of a person. As the musculoskeletal structure changes with age growth, a person's facial/head shape could change significantly over his/her different lifestages. Besides, aesthetic transformation and unique life experiences also stimulate significant physical changes like facial texture features, hairstyle, accessories, and motion behavior of a person, compounding the difficulties of learning effective representation to preserve identity. (2) **Limited Data Quantity and Quality for Each Lifestage.** In prior research, specific momentary data is required with either "standard inputs" (*i.e.,* high-quality front-view images for 3D-aware head generation/editing Rai et al. (2024)), or sufficient geometric cues (*i.e.,*multi-view capture systems Pan et al. (2024); Cheng et al. (2023); Kirschstein et al. (2023b); Wuu et al. (2022); Yang et al. (2020); Yu et al. (2020), depth sensors Livingstone & Russo (2018); Cosker et al. (2011), or short video sequences Zielonka et al. (2023); Gafni et al. (2021) for 3D/4D head avatar reconstruction). In contrast, it is infeasible to capture lifelong data of a person with sufficient geometric cues, or a unified frontal camera view for each lifestage. Oftentimes, one's lifetime is recorded through unstructured image collection, with extremely uneven data volume and viewpoints over different moments. Such data poses a significant challenge to high-fidelity 3D head avatar modeling in both appearance-realistic and geometry-plausible aspects. (3) **Explicit-controlled Animation in Full Scales.** Aside from lifelike reconstruction, a lifelong avatar is also expected to be controllable in terms of expression, shape, viewpoint, headpose, and appearance across a person's different age periods. Fueled by the disentangled space of parametric morphable head models like FLAME Li et al. (2017) and the expressiveness of neural field representation Mildenhall et al. (2020); Müller et al. (2022), previous arts of 3D head avatar animation Zheng et al. (2022a); Gafni et al. (2021); Zheng et al. (2022b); Zielonka et al. (2023); Qian et al. (2023); Kabadayi et al. (2023) could support head animation at different scales with vivid details. However, none of these methods could manipulate *moments of life*. How to learn a personalized disentangled space from highly unstructured data, that enables full-scale control without losing realism is yet unknown.

We present **TimeWalker** – a baseline solution that tackles the above challenges for lifelong head avatar modeling. To get invariant identity representation, and overcome the limited data issues, we track back to the principle of the classic 3DMMs, where a specific 3D mesh can be approximated by a mean template with shape/expression's main modes of variation in an additive combination manner. Analogously, TimeWalker models a person's each lifestage by the additive combination of a shared representation in canonical space, and a set of neural head basis. The former serves as the invariant, *i.e.,*average head representation, of the person's identity across different ages. The latter encodes the main modes of variation of moment-specific head attributes via a *Dynamic Neural Basis-Blending Module (Dynamo)*. The personalized space parameterized by these two components could provide both geometric and appearance priors to life moments with rare data. To achieve full-scale control while ensuring realism, we introduce *Dynamic 2D Gaussian Splatting (DNA-2DGS)* upon the additive combination framework. Specifically, we tailor Gaussian splatting representation to store the color, density, and correspondence of the heads via defining a set of Gaussian surfels in canonical space, and deforming them subsequently. To make the surfels animatable, we utilize two guidances to deform the surfels – the deformation fields driven from the neural head basis, and motion warping fields rooted from FLAME expression coefficients. In this way, the surfels could be animated properly

without losing geometric realism caused by FLAME's fixed topology, or explicit control problems caused by implicit neural head basis representation. With the above designs, TimeWalker can learn a well-disentangled personalized neural space (Fig. 1) only from one's lifelong unstructured image collection.

To enable modeling heads on the lifelong scale, a substantial dataset of the same individual at different time points is necessary. However, this requirement is challenging to fulfill with current open-source datasets since they neglect the lifelong human concept. Thus, we construct TimeWalker-1.0, a large-scale head dataset comprising 40 celebrities' lifelong image collection sourced from various Internet data. It contains over 2.7 million individual frames, with each identity consisting of $15K - 260K$ frames and diverse age and head pose distributions. In experiments, we show the ability of TimeWalker to reconstruct and animate avatars across decoupled dimensions with realistic rendering effects. Then, we demonstrate the superior rendering and geometry reconstruction quality by comparing our models to state-of-the-art momentary 3D head models. We also demonstrate the effectiveness of our designs with extensive ablation studies. Finally, we show the potential benefits of our model to downstream applications in 3D Editing.

## 2 RELATED WORKS

**Personalized Head Avatars.** 3DMM Blanz & Vetter (1999), as the foundational work in 3D human head modeling, constructs a generic *head* space via the linear combination of a mean template mesh and low-dimensional linear subspaces of shape and expression from PCA. Subsequent research extends 3DMM to personalized head mesh modeling. For instance, Chaudhuri et al. (2020) predicts personalized corrections on a 3DMM prior to obtain user-specific expression blendshapes and dynamic albedo maps. Zhu et al. (2023) learns personalized face details via multi-view image fusion from virtually rendered multi-view input images. Recent advancements have extended focus to creating animatable personal *head* avatars with realistic rendering. Methods like NerFace Gafni et al. (2021) and IM Avatar Zheng et al. (2022a) leverage FLAME Li et al. (2017) expression coefficients to drive neural scene representation networks, implicitly representing head avatars from monocular video inputs. INSTA Zielonka et al. (2023) enhances training speed and enables avatar control through a mesh-based warping field. GaussianAvatars Qian et al. (2023) and FlashAvatar Xiang et al. (2024) associate Gaussian points with a 3D parametric model and generate personalized head avatars through expression parameters as the condition to Gaussian offset. By harnessing the formidable generative prior, DreamBooth Ruiz et al. (2023) and Lora Hu et al. (2021) customize diffusion models from multiple images of a particular subject to produce personalized outcomes. However, these approaches are limited to static 2D results, lacking 3D consistency and animation capabilities. Other works like DiffusionAvatars Kirschstein et al. (2023a) and GANAvatar Kabadayi et al. (2023) utilize generative models to create personalized head avatars. The former fine-tunes ControlNet Zhang et al. (2023a) with NPHM Giebenhain et al. (2023) features. The later distill EG3D Chan et al. (2022) to single appearance. While these pipelines excel in constructing personalized head avatars, they focus on momentary representations, and none address the challenge of representing personalized spaces over a lifelong scale. Our pipeline takes a step in this direction with a foundational solution, enabling the construction of a lifelong replica. A comparison between TimeWalker and the representative methods mentioned above is shown in Tab. 1.

**Neural Representation for Static Reconstruction.** In contrast to traditional explicit reconstruction methods like meshes, point cloud and voxel grids, neural representation models, such as NeRF Mildenhall et al. (2021), show promise with high-fidelity rendering Barron et al. (2022), efficient training Müller et al. (2022); Yu et al. (2021a), and mobile deployment Chen et al. (2023). These models leverage differentiable rendering to refine parameters and minimize overfitting. Recent enhancements introduce explicit structures to boost rendering performance and training efficiency: InstantNGP Müller et al. (2022) adopts a multi-resolution hashgrid to streamline scene feature storage and expedite training. 3DGS Kerbl

| | Lifelong Replicas | Animation Expression | Shape | Mesh Reconstruction | High-Fidelity Rendering |
|---|---|---|---|---|---|
| Gaussian Surfels Dai et al. (2024) | ✗ | ✗ | ✗ | ✓ | ✓ |
| INSTA Zielonka et al. (2023) | ✗ | ✓ | ✗ | ✗ | ✓ |
| FlashAvatar Xiang et al. (2024) | ✗ | ✓ | ✗ | ✗ | ✓ |
| GANAvatar Kabadayi et al. (2023) | ✗ | ✓ | ✗ | ✗ | ✓ |
| **TimeWalker (ours)** | ✓ | ✓ | ✓ | ✓ | ✓ |

Table 1: **TimeWalker** enables preserving identity consistency in the long-horizon time spectrum (*Lifelong Replica*), with explicit-controlled animation at full scale (*Animation-Expression/Shape*). It also supports surface reconstruction and produces dynamic mesh efficiently under sparse view observations for each life stage (*Mesh Reconstruction*), without losing rendering realism (*High-Fidelity Rendering*). Better zoom in for details.

et al. (2023) uses explicit Gaussian Splatting for rendering, achieving fast inference rates (>100FPS) without network reliance. Gaussian Surfels Dai et al. (2024), an evolution of 3DGS, refines Gaussian kernels for depth inconsistency problems rooted in 3DGS, results in high-quality mesh reconstruction

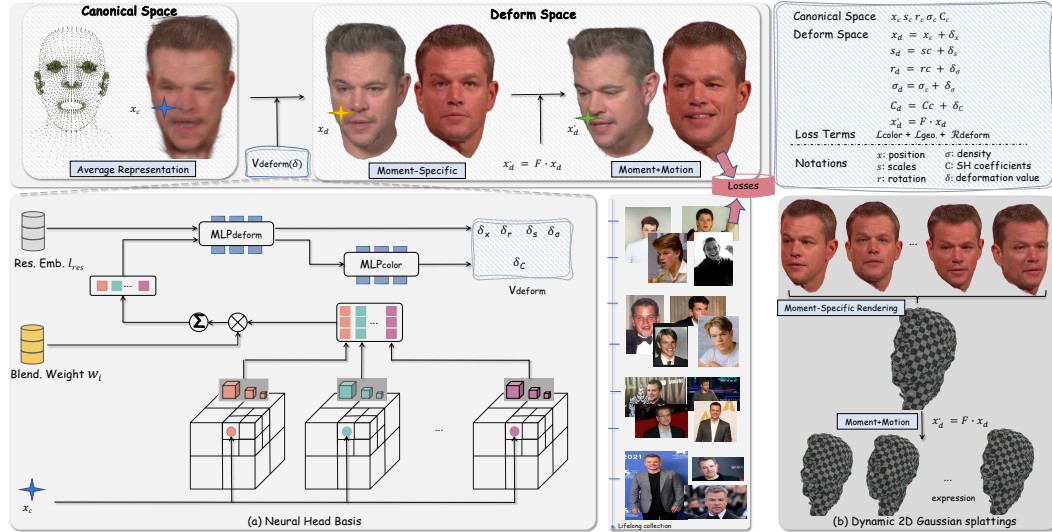

Figure 2: **Method Overview.** TimeWalker constructs a lifelong scale 3D avatar from unstructured photo collections spanning years, maintaining realism and animation fidelity. The model is rooted in the principle of linear combination space, and innovated into an interpretable, scalable, and steerable neural personalized feature space with two key components: Dynamic Neural Basis-Blending Module (Dynamo) and Dynamic 2D Gaussian Splatting (DNA-2DGS). Concretely, Gaussian Surfels initialized in canonical space using the FLAME template represent an individual's average head. Neural Head Basis deformations model moment-specific representations, while DNA-2DGS applies motion warping fields driven by FLAME parameters to capture expressions and movements, generating multi-dimensional head avatars (*i.e.,moment-motion* in deform space).

and realistic rendering, especially under sparse view conditions. Our work extends this representation to dynamic human head modeling, enabling effective dynamic avatar animation.

**Age Progressing Modeling.** Research on simulating aging effects has been prominent in recent decades. Methods like RFA Wang et al. (2016) and IAAP Kemelmacher-Shlizerman et al. (2014) have led the way in creating average faces and transferring texture differences between age groups to model aging. GAN-based approaches like S2GAN, and Face Aging GAN,He et al. (2019); Wang et al. (2018) generate subtle texture variations across different ages. Acknowledging the importance of shape and texture in age modeling, various techniques Lanitis et al. (2002); Suo et al. (2009; 2012); Yang et al. (2016) have emerged to address both simultaneously. Innovative diffusion-based text-to-video pipelines such as DreamMachine, Kling, and Gen3 AI; Kli (2024); Gen (2024) have showcased the ability to model age progression via hallucinating human-aging videos from textual cues, yielding impressive outcomes. However, these generative methods struggle to achieve explicit and comprehensive head animation (*e.g.,* expressions and shape variations), and face challenges in maintaining robust 3D consistency, limiting their functionality in creating personalized spaces.

## 3 TIMEWALKER

Our work targets to comprehensively construct a 3D head avatar of a person on a lifelong scale, as opposed to current trends Zielonka et al. (2023); Kabadayi et al. (2023); Qian et al. (2023); Zheng et al. (2022a); Xiang et al. (2024); Zheng et al. (2022b) that reconstruct and animate a person at the momentary level. This new setting introduces further puzzles to head avatar modeling – how to faithfully capture both shared and specific traits of the target person over different ages, while keeping the flexibility of animation in full scales (*i.e.,* facial expression, face shape)?

The primary challenge in constructing a lifelong head avatar lies in embedding the lifestage dimension during modeling. Changes to the head across different lifestages —such as variations in appearance, facial shape, and even accessories—are difficult to explicitly define. Moreover, these variations must be disentangled from other dimensions to allow for decoupled animation. This challenge is compounded by the often limited quality and quantity of data available for each lifestage, further complicating accurate modeling across a lifetime.

To address the challenges, we introduce a novel neural parametric model (see Fig. 2) that captures an average representation of a person's identity in canonical space, and extends to moment-specific head attributes through a set of Neural Head Bases. In Sec 3.1, we first introduce a preliminary 2D Gaussian representation Dai et al. (2024), which forms the foundational layer of our pipeline,

ensuring high-fidelity rendering and dense meshing for the base frame, even with limited camera views across lifestages. In Sec. 3.2, we detail how the Personalized Neural Parametric Model constructs a personalized neural feature space via a Linear Combination Space formulation. The model leverages a Linear Combination Space formulation to handle the complexities introduced by different lifestages through the Neural Head Basis, while maintaining a shared canonical representation. Moving forward, we explore the geometric representation behind the Personalized Neural Parametric Model in Sec. 3.3– how our DNA-2DGS, a dynamic extension of 2DGS, fosters rendering and drives the dense mesh with different motion signals. In Sec. 3.4, we introduce training process designs, and discuss how we build a lifelong personalized space with disentangled control along different dimensions in Sec. 3.5.

## 3.1 PRELIMINARIES

Extended from Gaussian Splatting Kerbl et al. (2023), Gaussian Surfels (2DGS) reduces one dimension and transforms the Gaussian ellipsoids into Gaussian ellipses. Specifically, a scene is depicted by a set of unconstructed Gaussian kernels with attribute $\{\mathbf{x}_i, \mathbf{r}_i, \mathbf{s}_i, \sigma_i, \mathcal{C}_i\}_{i \in \mathcal{P}}$, where $i$ is the index of each Gaussian kernel, and $\mathbf{x}_i \in \mathbb{R}^3$, $\mathbf{r}_i \in \mathbb{R}^4$, $\mathbf{s}_i \in \mathbb{R}^3$, $\sigma_i \in \mathbb{R}$ and $\mathcal{C}_i \in \mathbb{R}^k$ respectively denotes the center position/rotation/opacity/spherical harmonic coefficients of each Gaussian's kernel. 2D Gaussian could be obtained by flatten the 3D Gaussian's rotation on $z$-axis (*i.e.*,$\mathbf{s}_i = [\mathbf{s}_i^x, \mathbf{s}_i^y, 0]^\top$). Given Gaussian distribution modeled as $G(\mathbf{x}; \mathbf{x}_i, \mathbf{\Sigma}_i) = \exp\left\{-0.5(\mathbf{x} - \mathbf{x}_i)^\top \mathbf{\Sigma}_i^{-1}(\mathbf{x} - \mathbf{x}_i)\right\}$, where $\mathbf{\Sigma}_i$ is the covariance matrix that can be unfolded as $\mathbf{R}(\mathbf{r}_i)\mathbf{S}_i\mathbf{S}_i^\top\mathbf{R}(\mathbf{r}_i)^\top$, with a scaling matrix $\mathbf{S}$ and a rotation matrix $\mathbf{R}(\mathbf{r}_i)$. $\mathbf{\Sigma}_i$ defines how the Gaussian surfel is stretched or compressed along different axes. Under the 2D Gaussian framework, the covariance matrix can be formed as $\mathbf{\Sigma}_i = \mathbf{R}(\mathbf{r}_i)\text{Diag}\left[(\mathbf{s}_i^x)^2, (\mathbf{s}_i^y)^2, 0\right]\mathbf{R}(\mathbf{r}_i)^\top$. The $\text{Diag}[\cdot]$ indicates a diagonal matrix. Practically, by blending all Gaussian kernels in the scene with depth-ordered rasterization, the color $\tilde{C}$, the normal value $\tilde{N}$, and the depth value $\tilde{D}$ of a pixel can be obtained by

$$\tilde{C} = \sum_{i=0}^{n} T_i \alpha_i \mathbf{c}_i, \quad \tilde{N} = \frac{1}{1 - T_{n+1}}\sum_{i=0}^{n} T_i \alpha_i \mathbf{R}_i[:, 2], \quad \tilde{D} = \frac{1}{1 - T_{n+1}}\sum_{i=0}^{n} T_i \alpha_i d_i(\mathbf{u}), \quad (1)$$

where $\alpha_i$ represents alpha-blending weight, and $1/(1 - T_{n+1})$ is a normalization scale for blending weight $T_i \alpha_i$, which is calculated with $T_i = \prod_{j=0}^{i-1}(1 - \alpha_j)$. $d_i(\mathbf{u})$ represents the adjusted depth value of the center of the Gaussian kernel. For the equation of normal, with the degenerated 2D ellipse, the normal direction of a Gaussian kernel can be directly extracted as $\mathbf{n}_i = \mathbf{R}(\mathbf{r}_i)[:, 2]$. Based on the above formula, Gaussian Surfels are surface-conforming primitives that project directly onto the image plane, ensuring precise local depth and normal blending processes. In contrast, 3DGS uses volumetric Gaussians, which can blur depth boundaries and reduce precision, especially in areas with complex or discontinuous surfaces. Consequently, Gaussian Surfels exhibits remarkable performance even in sparse view settings, a crucial requirement for our intricate pipeline.[1]

In this work, we represent human head upon Gaussian Surfels, taking full advantage of its excellent reconstruction performance without sacrificing the realism of the rendering. Notably, 2DGS, built for static reconstruction, would encounter a significant challenge in generating dynamic mesh sequences, due to the time-consuming nature of post Poisson Meshing procedure Kazhdan & Hoppe (2013). Thus, we devise a Defer Warping strategy upon the representation (*i.e.,* DNA-2DGS in the framework) to surmount this limitation. The specifics are thoroughly outlined in Sec. 3.3.

## 3.2 PERSONALIZED NEURAL PARAMETRIC MODEL

### 3.2.1 NEURAL LINEAR COMBINATION SPACE

In the quest to pioneer lifelong personalized animation spaces, our primary objective is to enable precise and controlled animation of individuals on a comprehensive scale. Leveraging linear combination proves to be an optimal strategy in this context, as it offers flexibility and universality in constructing compact representations through the combination of base vectors that encapsulate core variations. It provides the theoretical foundation for constructing a wide range of shapes, textures, and expressions even from limited data. Additionally, it simplifies optimization and computation, ensuring consistent and predictable results. This concept traces its origins to traditional 3D Head Morphable

---

[1]More theoretical details about Gaussian Surfels and qualitative as well as quantitative comparison of mesh reconstruction between 3DGS and 2DGS can be referred to Dai et al. (2024).

Models, where a head is modeled by linearly combining expression and shape parameters with their corresponding basis functions: $\mathbf{M}(\alpha, \beta, \gamma) = \left( \bar{S} + \sum_{i=1}^{N_s} \alpha_i \mathbf{s}_i, \ \bar{T} + \sum_{i=1}^{N_t} \beta_i \mathbf{t}_i, \ \bar{E} + \sum_{i=1}^{N_e} \gamma_i \mathbf{e}_i \right)^2$ Building upon this foundation, we extend the concept to a neural linear combination space, integrating multiple animation dimensions in an additive, learnable, and scalable manner, as formulated in Eq 4. Particularly, as illustrated in Fig 2, to characterize the average head representation of individuals, we initialize a set of Gaussian Surfels in canonical space based on FLAME template. With linear addition of the deformation value produced from Neural Head Basis (detailed in Sec 3.2.2), the canonical Gaussian Surfels could be deformed from the average representation to a specific lifestage (*i.e.,* moment-specific representation in the Figure). This process facilitates the modeling of a neutral head at any life moment, and moreover, within our framework, this is achieved without the need for direct supervision. Sequentially, to enable motion-based modeling (*e.g.,* expression changes) and dynamic avatar animation, we further utilize motion warping fields rooted from FLAME parameters (Sec 3.3.1), to drive the moment-motion modeling. Along with the analysis-by-synthesis training process (Sec 3.4), these designs allow us to create multi-dimensional realistic head avatars in a breeze.

### 3.2.2 Neural Head Basis

With the assumption that any lifestage of one person can be linearly blended from his/her several key characteristic variations across lifestages, we now introduce how to capture and learn these variations within the linear neural feature space from the design of the Neural Head Basis, and how to obtain a *compact* set of the bases from our *Dynamo* module. Suppose the number of head basis is $N$, for a point $\mathbf{x}_c$ picked from the canonical space, its feature at a specific lifestage could be formed as a linear combination of $N$ neural head basis $\mathcal{H}$: $\mathbf{f}(\mathbf{x}_c) = \sum_{i=1}^{N} \omega_i \mathcal{H}_i(\mathbf{x}_c)$, where $\omega_i$ is the learnable blending weight for each neural head basis. To store the features compactly, we utilize multi-resolution Hashgrid Müller et al. (2022); Kirschstein et al. (2023b), a hashmap-based cubic structure that enables the learnable features stored in a condensed form. When querying features for $\mathbf{x}_c$, the hashgrid looks up nearby features at various scales $J$, and cubic linear interpolation is applied to determine the final feature corresponding to the location. Thus, for each $H_i$, it is constructed by $\mathcal{H}_i = \text{C-LinearInterp}\left( \{ h_j^i \}_{j=1}^{J} \right)$.

**Dynamic Neural Basis-Blending Module (Dynamo).** *How to set the number of head basis?* One intuitive way would be – assigning a fixed amount that is equivalent to the number of appearances in the character's data, with each basis independently learning the character's features for a specific lifestage. For instance, to build a comprehensive personal space for Leonardo, encompassing ten distinct lifestages, we could initiate ten individual neural bases. Each of these bases is specifically tasked with mastering the intricacies of a single lifestage. However, this idea is inefficient and redundant in the feature space, as a person's appearance evolves over their lifetime, their core characteristics that define their identity remain fundamentally consistent. Even with temporary alterations like heavy makeup or accessories, we can still recognize individuals by their underlying features. Besides, we expect the feature space could be interpretable, scalable, and controllable. Thus, our goal for the neural basis is to capture these deeper characteristics rather than solely memorizing superficial appearances. To this end, we introduce *Dynamo* to dynamically adjust the number of bases during the learning process of blending weight and hashgrids. We begin by initializing a set of learnable blending weights $\{\omega\}_{i=1}^{N}$ and grids $\mathcal{H}$, which align with the number of lifestages of the target person. Throughout the learning process, if $\{Indicator(\omega_i^k < \kappa) = 1\}$ consistently reveals that a hashgrid's weight falls below lower a preset threshold $\kappa$ across modeling multiple lifestages $k$ at an iteration $Q$ with sampled iteration interval $q$, this signals that this grid is not effectively learning meaningful features. In response, we deactivate that hashgrid. By the end of the training process, this pruning strategy ensures that each remaining Neural Head Basis is efficiently capturing core, identity-defining features across lifestages. Additionally, it reduces memory requirements for storing these features, leading to a more efficient and scalable representation.

**Residual Embedding.** Using *Dynamo* outlined above, we obtain a collection of feature embeddings that compactly capture the subject's moment-specific attribute. As global compensation for each lifestage, we introduce a set of residual embedding $\mathbf{l_{res}}$, which are concatenated with the blended features $\mathbf{f}(\mathbf{x}_c)$. The concatenated features are then forwarded into a MLP$_{\text{deform}}$ to derive position $\mathbf{x}$, rotation $\mathbf{r}$, scale $\mathbf{s}$, and opacity $\sigma$ deformations of the Gaussian kernel, as well as another feature vector

---

²Where $M(\alpha, \beta, \gamma)$ is the full model (shape, texture, expression), $\bar{S}, \bar{T}, \bar{E}$ are the mean shape, texture, and expression, $\mathbf{s}_i, \mathbf{t}_i, \mathbf{e}_i$ are the basis vectors, $\alpha_i, \beta_i, \gamma_i$ are the corresponding coefficients, and $N_s, N_t, N_e$ are the number of components for shape, texture, and expression, respectively. Please refer to the original paper for more details Blanz & Vetter (1999); Li et al. (2017).

which is subsequently passed through a $\text{MLP}_{\text{color}}$ to generate the SH coefficients $\mathcal{C}$ deformation:

$$\delta_{\mathbf{x}}, \delta_{\mathbf{r}}, \delta_{\mathbf{s}}, \delta_{\sigma}, \mathbf{f}_{\text{deform}}(\mathbf{x}_c) = \text{MLP}_{\text{deform}}(\mathbf{f}(\mathbf{x}_c), \mathbf{l}_{\mathbf{res}}); \delta_{\mathbf{C}} = \text{MLP}_{\text{color}}(\mathbf{f}_{\text{deform}}(\mathbf{x}_c)). \quad (2)$$

The network learns the deformations $\delta$ of Gaussian attributes, which are then additively combined with the Gaussian average in canonical space. It yields the character's moment-specific features:

$$[\mathbf{x}_d, \mathbf{r}_d, \mathbf{s}_d, \sigma_d, \mathbf{C}_d] = [\mathbf{x}_c, \mathbf{r}_c, \mathbf{s}_c, \sigma_c, \mathbf{C}_c] + [\delta_{\mathbf{x}}, \delta_{\mathbf{r}}, \delta_{\mathbf{s}}, \delta_\sigma, \delta_{\mathbf{C}}]. \quad (3)$$

The whole combination process can be summarized below:

$$[(\mathbf{x}, \mathbf{r}, \mathbf{s}, \sigma)_d, \mathbf{C}_d] = \underbrace{[(\mathbf{x}, \mathbf{r}, \mathbf{s}, \sigma)_c, \mathbf{C}_c]}_{\text{Mean Representation}} + \underbrace{[\text{MLP}_{\text{d}}(\sum_{i=1}^{N} \omega_i \mathcal{H}_i(\mathbf{x}_c), \mathbf{l}_{\mathbf{r}}), \text{MLP}_{\text{c}}(\text{MLP}_{\text{d}}(\sum_{i=1}^{N} \omega_i \mathcal{H}_i(\mathbf{x}_c), \mathbf{l}_{\mathbf{r}}))]}_{\text{Neural Linear Additive Deformation}}.$$

$$(4)$$

### 3.3 DYNAMIC 2D GAUSSIAN SPLATTINGS (DNA-2DGS)

The question now is – given a motion target/reference, how can we effectively drive the moment-specific representation to capture motion dynamics with precision, controllability and realism? This requires further deformation of the Gaussian Surfels to reflect various motion-related changes like expressions or shapes. Thus, we introduce *DNA-2DGS*, which tackles the problem from both rendering (the upper area of Fig 3) and meshing (the lower area of Fig 3) aspects.

#### 3.3.1 DYNAMIC GAUSSIAN RENDERING

To realize animation, one intuitive strategy is to incorporate an additional MLP-based warping field. In our experimental setup, we implemented this component as a naive dynamic version of Gaussian Surfels [3]. While this method shows proficiency in handling deformations induced by various expressions, it falls short in capturing the entire range of human head motions, particularly around eyes. Moreover, disentangling the animation of appearance and expression remains challenging when both components of the warping field are designed similarly. In contrast, we integrate motion warping fields inspired by INSTA Zielonka et al. (2023). During the preprocessing, we acquire a tracked mesh $M^{def}$ through FLAME fitting and a mean template $M^{canon}$ defined within a canonical space, both sharing identical topology. Unlike INSTA which utilizes the warping field to inversely project points from deformed space back to canonical space, we define a deformation gradient $\mathbf{F} \in \mathbb{R}^{4 \times 4}$ in the form of a transformation matrix. This matrix projects Gaussian Surfels

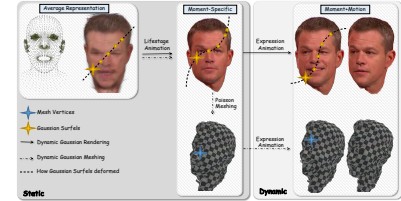

Figure 3: **Dynamic Gaussian Rendering & Meshing.** For the former, Gaussian Surfels are firstly animated with lifestage and expression respectively, followed by rasterization to produce high-fidelity results. For the latter, after deformed with lifestage, the Poisson Meshing process is used to obtain a dense mesh. Then, expression animation is performed on the mesh vertices. Better zoom in for details.

$\mathbf{x}_d$ from the moment-specific static space to the dynamic motion space. Concretely, for each Gaussian Surfel $\mathbf{x}_d$, we employ a nearest triangle search algorithm to compute $\mathbf{F} = \mathbf{L}_{def} \cdot \Lambda^{-1} \cdot \mathbf{L}_{canon}^{-1}$, where $\mathbf{L}_{canon}$ and $\mathbf{L}_{def}$ is Frenet coordinate system frames, and $\Lambda$ is a diagonal matrix that takes scaling factor into account.[4] With computed $\mathbf{F}$ Gaussian Surfels are further deformed $\mathbf{x}_d' = \mathbf{F} \cdot \mathbf{x}_d$, guided towards specific movements. This process enables translation of expression/shape signals into animated outputs. The double-deformed Gaussian surfels are then rasterized to produce final renderings.

#### 3.3.2 DYNAMIC GAUSSIAN MESHING

Current Gaussian Surfels Dai et al. (2024) enables high-quality surface reconstructions after training via Gaussian point cutting and Poisson Mshing Kazhdan & Hoppe (2013), using extracted data from rendering results. However, this method is primarily suited for *static* settings, limiting its applicability to dynamic head avatars. Additionally, the time-intensive nature of Poisson reconstruction process makes it impractical for mesh sequence reconstruction. To address these challenges, we

---

[3]Please refer to the results of Gaussian-Surfel++ in Appendix A.5.2.
[4]Please refer to INSTA Zielonka et al. (2023) for more theoretical details.

introduce *Defer-Warping*, an adapted Gaussian surface reconstruction strategy tailored specifically for dynamic head reconstruction. Specifically, unlike standard rendering & meshing processes that apply deformations before meshing, we delay the motion animation after the Poisson meshing process of moment-specific representation. This allows us to render results under moment-specific conditions customized to every single lifestage, effectively eliminating motion-induced artifacts (such as artifacts on mouth regions caused by talking) and generating appearance-specific static meshes. After Poisson meshing of the static result, the delayed motion animation generates dynamic mesh sequences by directly manipulating the vertices of reconstructed mesh. This process faithfully captures motion-driven deformations. Refer to Appendix A.5.3 for a detailed comparison of mesh generation.

## 3.4 TRAINING

We apply the end-to-end training manner that enables the simultaneous optimization of the explicit Gaussian surfels, multiple hashgrid, feature latent, and implicit deform MLP & color MLP. Before formal training, we introduce a warm-up phase where we suspend the optimization of the Neural Head Basis. This step aims to guide the Gaussian surfels in canonical space towards a mean representation (Detailed in A.3 of Appendix). For the Gaussian Splatting, we follow the densify and pruning strategies of 3DGS to adaptively adjust the number of Gaussians. To guide the optimization of the whole system, our total loss $\mathcal{L}_{\text{total}}$ consists of three parts: (1) Image Level Supervision. Similar to 3DGS Kerbl et al. (2023), This term includes photometric $L1$ loss $\mathcal{L}_{\text{rgb}}$ and ssim loss $\mathcal{L}_{\text{ssim}}$. An additional perceptual loss $\mathcal{L}_{\text{lpips}}$ Johnson et al. (2016) with AlexNet encoder Krizhevsky et al. (2012) is included to improve the rendering quality. (2) Geometry Level Supervision. Inspired by INSTA Zielonka et al. (2023), we include $\mathcal{L}_{\text{depth}}$ to enforce a better Gaussian geometry based on FLAME tracked mesh. Specifically, we apply $L1$ loss between the predicted depth from Eq. 1 and GT depth rasterized from FLAME mesh, with respect to a specific face region segmented by a ready-to-use face parsing model Yu et al. (2021b). Following Guassian Surfels Dai et al. (2024), we apply both $\mathcal{L}_{\text{normal}}$ and $\mathcal{L}_{\text{consist.}}$. The former acts as a prior-based supervision to improve the training stability, and the latter enforces consistency between the rendered depth $\tilde{D}$ and rendered normal $\tilde{N}$:

$$\mathcal{L}_{\text{normal}} = 0.04 \cdot (1 - \tilde{\mathbf{N}} \cdot \hat{\mathbf{N}}) + 0.005 \cdot L_1(\nabla \tilde{\mathbf{N}}, \mathbf{0}), \mathcal{L}_{\text{consist.}} = 1 - \tilde{\mathbf{N}} \cdot N(V(\tilde{\mathbf{D}})), \quad (5)$$

where $\hat{\mathbf{N}}$ denotes the normal map from a pretrained monocular model from Eftekhar et al. (2021) and $\nabla \tilde{\mathbf{N}}$ represents the gradient of the rendered normal. (3) Regulation. To ensure that the Gaussian attributes do not deviate significantly from their mean representation, we employ a $L1$ regulation to the deformation of the Gaussian attributes $[\delta_{\mathbf{x}}, \delta_{\mathbf{r}}, \delta_{\mathbf{s}}, \delta_{\sigma}, \delta_{\mathbf{C}}]$ and penalize large deformation. The total loss function can be constructed as:

$$\mathcal{L}_{\text{total}} = \underbrace{\lambda_{\text{r}}\mathcal{L}_{\text{rgb}} + \lambda_{\text{s}}\mathcal{L}_{\text{ssim}} + \lambda_{\text{l}}\mathcal{L}_{\text{lpips}}}_{\text{Image Level Supervision}} + \underbrace{\lambda_{\text{d}}\mathcal{L}_{\text{depth}} + \lambda_{\text{n}}\mathcal{L}_{\text{normal}} + \lambda_{\text{c}}\mathcal{L}_{\text{consist.}}}_{\text{Geometry Level Supervision}} + \underbrace{\lambda_{\text{reg}}\mathcal{L}_{\text{deform}}}_{\text{Regulation}} \quad (6)$$

where $\lambda_{\text{r}}$ denotes dynamic weight based on a face parsing mask. Note that, for the warm-up phase, as we only optimize the attributes of Gaussian kernel, we do not include regulation terms. For the formal training phase, all loss terms are employed. *Refer to Appendix Tab. 3 for details of hyperparameters.*

## 3.5 BUILDING A LIFE LONG PERSONALIZED SPACE

*How does TimeWalker construct a lifelong personalized space?* The Gaussian Surfels in canonical space that characterize the individual average representation, are additively combined with moment-specific head attribute representations driven from a set of Neural Head Basis to span a moment-specific head avatar. For each moment-specific head avatar, we further warp the Gaussian Surfels via expression or shape signals to get the motion-specific head performing. By separating the moment-specific deformation from the motion-specific warping, we are able to decouple the driving of the head in multiple dimensions - lifestage, expression, shape, novel view, and *etc*, - constructing a comprehensive, steerable personalized space. Please refer to Sec. A.2 in Appendix for details.

*What are the core benefits of learned space?* Despite being trained on a restricted[5] dataset, our model demonstrates: (1) the ability to generate a comprehensive personalized space with full-scale animation (Fig. 1), steerable reenactment (Sec. 4.1), and high-fidelity outcomes (Sec. A.2). (2) In contrast to alternative generative or animatable avatar techniques, our approach offers superior disentanglement of properties and ensures consistency across animated dimensions in terms of both appearance (Sec. A.5.2) and geometry (Sec. A.5.3). (3) With faithful consistency, our method could potentially improve the quality of downstream tasks (Sec. A.6).

---

[5]The *restricted* here refer to the limited data quantity and quality for each lifestage. For example, the side-profile data is very sparse/limited.

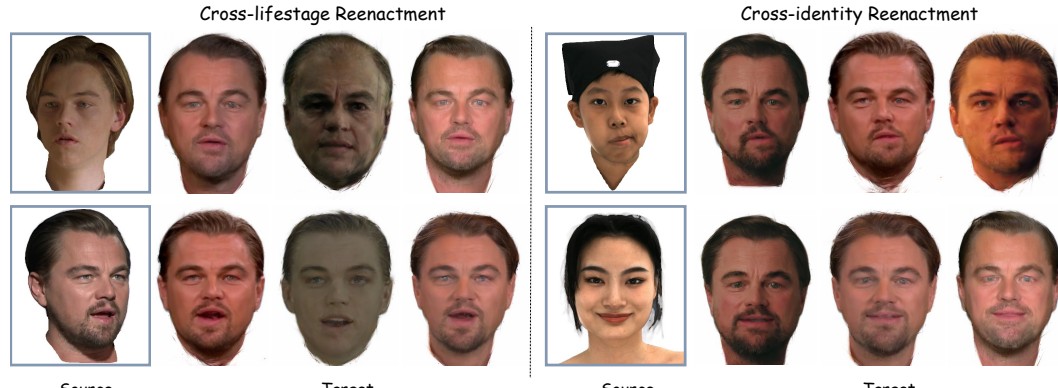

Figure 4: **Reenactment.** We demonstrate the cross-lifestage reenactment (self-reenacment) with TimeWalker-1.0 and cross-identity reenactment with RenderMe-360 Pan et al. (2024), in Leonardo personalized space.

Table 2: **Ablation study.** Pink indicates the best and orange indicates the second.

| | **Loss Term** | | | **Dynamo Module** | | | |
|---|---|---|---|---|---|---|---|
| Item | w/o $\mathcal{L}_{\text{lpips}}$ | w/o $\mathcal{L}_{\text{geometry}}$ | w/o $\mathcal{L}_{\text{deform}}$ | w/o Dynamo | w/ 1 hashgrid | w/ all hashgrid | **Ours** |
| PSNR↑ | 26.56 | 27.25 | 24.79 | 21.69 | 24.84 | 26.86 | 27.20 |
| SSIM↑ | 0.916 | 0.943 | 0.886 | 0.767 | 0.890 | 0.938 | 0.941 |
| LPIPS↓ | 0.18 | 0.080 | 0.165 | 0.197 | 0.119 | 0.078 | 0.077 |

## 4 EXPERIMENTS

*Please refer to Appendix for comparisons with SOTAs( A.5.2), personalized space visualization( A.5.1) and mesh comparison( A.5.3). In main paper, we unfold our method with reenactments and ablations.*

### 4.1 DATASETS

To fully evaluate our pipeline, we construct a large-scale head dataset, TimeWalker-1.0, featuring lifelong photo collections of 40 celebrities. The data volume ranges from $15K$ to $260K$ per celebrity, capturing diverse variations across life stages. For details and statistics of the dataset, as well as comparisons with other datasets, refer to Appendix A.4. We also evaluate on **INSTA** Zielonka et al. (2023), **Nersemble** Kirschstein et al. (2023b), and **RenderMe-360** Pan et al. (2024) for further comparison and cross-identity reenactment.

### 4.2 REENACTMENT

Fig. 4 shows expression animation with two types of reenactments. The *cross-lifestage reenactment* on the left showcases how head avatars from different lifestages can consistently perform the same expression, animated from a source avatar belonging to a different time period. The *cross-identity reenactment* on the right shows Leonardo in different lifestages are driven by unseen novel expression from RenderMe-360 Pan et al. (2024). The rendering result shows that multiple head avatars generated by the same personal space from TimeWalker are able to extrapolate novel expressions.

### 4.3 ABLATION STUDIES

To validate the effectiveness of our method components, we conduct several ablation experiments in terms of our Dynamo design and loss terms. All the ablation experiments are conducted on 3 individuals with respective 9, 10 and 13 lifestages. We keep other settings unchanged except the ablation term. The results are demonstrated in Tab.2.

**Loss Term.** We evaluate the effectiveness of each loss-term design, including $\mathcal{L}_{\text{lpips}}$, $\mathcal{L}_{\text{geometry}}$ and $\mathcal{L}_{\text{deform}}$. As shown in Tab. 2, the performance significantly drops without $\mathcal{L}_{\text{lpips}}$ or $\mathcal{L}_{\text{deform}}$, and $\mathcal{L}_{\text{geometry}}$ does not contribute to the rendering result. This is consistent with the ablation experiments in Dai et al. (2024), as the geometry-based loss mainly contributes to the high quality mesh reconstruction rather than the realism of avatar rendering. In Fig. 13 we also visualize the mesh without $\mathcal{L}_{\text{geometry}}$. As shown in Fig. 5, $(a)$ including $\mathcal{L}_{\text{lpips}}$ during training helps reduce smoothness in high frequency parts like hair and beard. $(b)$ $\mathcal{L}_{\text{geometry}}$ does not contribute to rendering result, but aids in mesh reconstruction. $(c)$ Training without $\mathcal{L}_{\text{deform}}$ results in strip-shaped Gaussian ellipses or even generation failed.

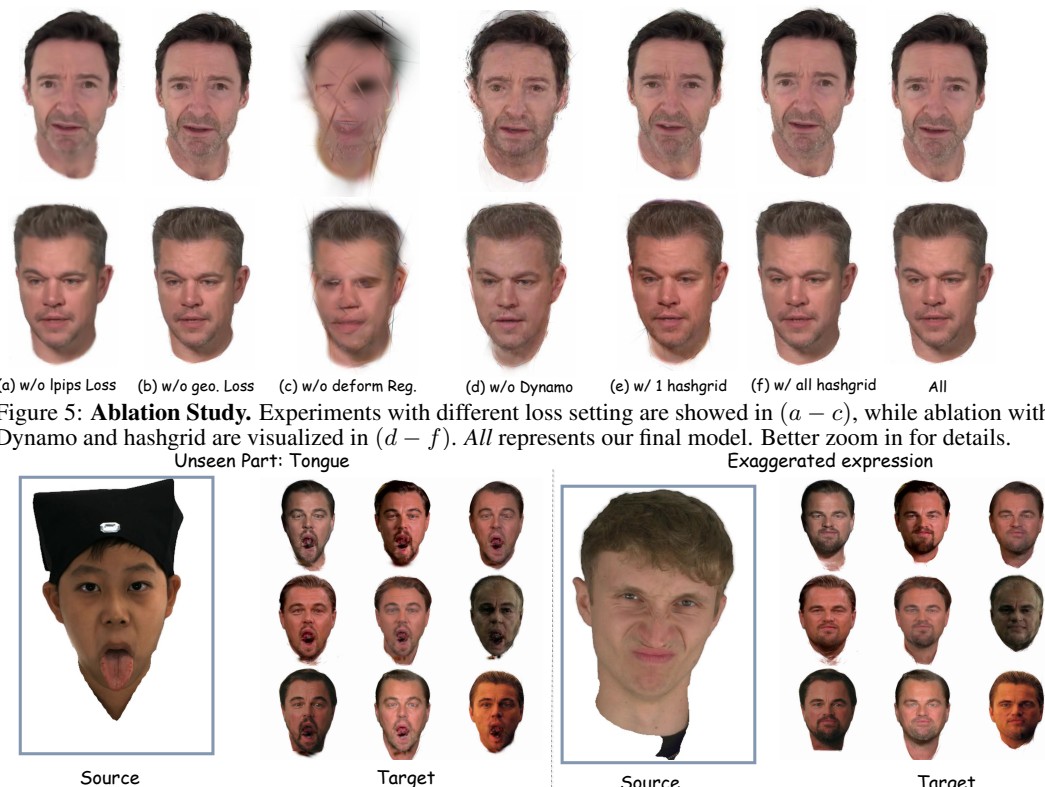

(a) w/o lpips Loss    (b) w/o geo. Loss    (c) w/o deform Reg.    (d) w/o Dynamo    (e) w/ 1 hashgrid    (f) w/ all hashgrid    All

Figure 5: **Ablation Study.** Experiments with different loss setting are showed in $(a - c)$, while ablation with Dynamo and hashgrid are visualized in $(d - f)$. *All* represents our final model. Better zoom in for details.

Figure 6: **Limitations.** Our method fails to model unseen parts like tongue during animation, and performs moderate expressions when driven by exaggerated expressions. (Source from Renderme-360 Pan et al. (2024) and NerSemble Kirschstein et al. (2023b))

**Dynamo.** We conduct ablation experiments on Dynamo in the Neural Head Basis, a key contribution that enables our pipeline to preserve the moment-specific attributes of individuals. Two protocols are used: (1) removing Dynamo and replacing it with the position $\mathbf{x}_c$ of each Gaussian surfels; (2) predefining and fixing the number of hashgrids (either one or matching the number of life stages). The quantitative result in Tab. 2 shows the necessity of Dynamo with multiple hashgrids, as reducing the hashgrid number or removing Dynamo leads to performance dropout. Interestingly, our pipeline with adaptive hashgrid number performs slightly better than the setting with all hashgrid, which demonstrates effectiveness of our hashgrid adaptation strategy. Fig. 5 also shows that, without Dynamo, the model tends to preserve moment-specific attributes in the learnable latent, leading to artifacts surrounding the head avatar ($(d)$ in Figure). Models trained with only one hashgrid cannot provide enough details on head surface and have obvious artifacts around the eyes ($(e)$), while those trained with adequate hashgrids perform similarly to our pipeline but require larger model sizes ($(f)$).

## 5    DISCUSSION

**Limitations.** There are two main limitations of our pipeline. First, due to the underlying FLAME base, our personalized avatar fails to capture exaggerated expressions and model unseen parts like the tongue during the animation, resulting in noticeable artifacts around the mouth area (Fig. 6). Second, due to data constraints, our study only provides a preliminary exploration toward lifelong perspective, rather than encompassing the entire life span from infancy to elderly. We aspire for our research to establish a baseline in this field, inspiring more effective approaches towards achieving comprehensive lifelong personalized space construction. A promising future direction is to leverage the prior knowledge of head structures from pre-trained generative models.

**Conclusion.** In this work, we present TimeWalker, a baseline solution for constructing personalized spaces that maintain long-term identity consistency while enabling explicit, full-scale animation control. Rooted in the additive combination principles of classic 3DMM, our approach innovates through a neural feature-based design with two key components: a Dynamic Neural Basis-Blending model to represent the head variations in a compact manner and a Dynamic 2D Gaussian Splatting module to construct dynamic dense head mesh. Our method makes it possible to create personalized spaces that evolve faithfully over a lifetime.

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

# A    APPENDIX

This appendix aims to enhance the discussion presented in our main paper, and demonstrate more details as supplementary. Firstly, we introduce additional related works as background supplementary in Sec. A.1. Then, we provide more details about the construction of a lifelong personalized space in Sec. A.2. We delve into the specifics of our implementation and training process in Sec. A.3. Following that, in Sec. A.4, we elaborate on the creation of our dataset, TimeWalker-1.0, detailing its construction, statistics, and a comparative analysis with existing datasets. Subsequently, in Sec. A.5, we present additional experiments, such as results in personalized spaces, comparisons with state-of-the-art techniques concerning 2D rendering and 3D mesh outputs, and more individual visualization. We showcase the outcomes of our downstream application, 3D editing, in Sec. A.6. Lastly, we summarize the broader impact of our work in Sec. A.7.

## A.1    ADDITIONAL RELATED WORK

Since our approach constructs the 3D personalized space from in-the-wild unstructured photos, we also discuss related developments in the research field of general 3D reconstruction from unstructured photos.

**3D Reconstruction from Unstructured Photo.** Reconstruction from Internet photo collections has been a long-standing topic in computer vision and computer graphics. Thought-provoking research such as Photo Tourism Snavely et al. (2006), Skeletal Sets Snavely et al. (2008), and Building Rome in a Day Agarwal et al. (2009) show great potential for applying structure from motion (SfM) algorithms on unstructured photo collections. Upon these pioneer works, multi-view stereo (MVS) algorithms Curless et al. (2010); Schönberger et al. (2016) and appearance modeling Kim et al. (2016) are proposed to improve the reconstruction quality. More recently, several works Martin-Brualla et al. (2021); Sun et al. (2022) model the scene by grafting these ideas into the neural radiance fields Mildenhall et al. (2020). There are also a series of great works focused on reconstructing heads or heads from Internet photos. For example, in Kemelmacher-Shlizerman (2013), the author holds the premise of deriving a 3D head shape basis directly from a large amount of Internet collection, and proposes to reconstruct an arbitrary 3D head from a single view image based on the shape basis. The subsequent work such as Kemelmacher-Shlizerman & Seitz (2011) focuses on recovering the head from personal photo collections, and Liang et al. (2016) aims to recover the personalized head. Still and all, not much attention has been paid to reconstructing human heads on a lifelong scale. The most related work to our project is PersonNeRF ( Weng et al. (2023)), which is driven from NeRF-W Martin-Brualla et al. (2021) to model personalize space for the human body across several years' data. However, this method assumes a person's body shape is roughly identical across years, which limits its scalability to lifelong settings. Beyond that, human head modeling is more challenging than body in terms of high-fidelity details and fine-grained capturing like subtle expression.

## A.2    BUILDING A LIFE LONG PERSONALIZED SPACE

In Sec. 3.5 of the main paper, we introduce our approach for constructing a personalized space with full-scale animation in a disentangled manner. In this section, we further elaborate on how we realize the animation of each dimension within this personalized space.

*Lifestage.* During training, our pipeline learns different blending weights $\{\omega\}_{i=1}^N$ for data in different lifestages. After training, we can adjust these weights to drive the lifestage in a disentangled manner. Fig. 10 illustrates the appearance diversity of individuals as they progress through different lifestages. This demonstrates the effectiveness of our pipeline in capturing a person's identity across different moments in their life.

*Expression.* To achieve expression and shape changes of the character while maintaining a consistent appearance, we use a motion warping field inspired by INSTA Zielonka et al. (2023). By manipulating expression parameters, we can update both the tracked mesh and the transformation matrix that maps from canonical space to deformation space. This enables us to achieve the desired expression-based warping.

*Shape.* As the FLAME mesh can be driven by expression and shape parameters in a disentangled manner, our head avatar can also be animated by shape with the same approach as expression.

*Novel view.* The Gaussian Splatting, as a type of 3D representation, can be rendered with arbitrary camera pose.

### A.3 IMPLEMENTATION DETAILS

We apply the end-to-end training manner with Gaussian Surfels as our basis representation. Instead of SfM Schönberger & Frahm (2016) based initializing in 3DGS Kerbl et al. (2023) and Gaussian Surfels Dai et al. (2024), we leverage the FLAME template and initialize the Gaussian kernel on its surface, with one Gaussian kernel on the center of each triangle face. The initialization of other attributes follows the original 3DGS implementation. All the components, including Neural Head Basis, Residual embedding, and deformation networks, join to start the end-to-end formal training for 30000 iterations. Please refer to Tab. 3 for detailed hyperparameters of TimeWalker setting during the training process. We follow Wu et al. (2023a) and apply the same initial learning rate and rate scheduler for all network components and Gaussian attributes. We use a single NVIDIA A100 GPU to train the model, and it costs $3 \sim 4$ hours on average for the whole training process.

**Warm Up.** Prior to formal training, we incorporate a warm-up phase wherein we utilize a person's data across all his/her lifestages to individually optimize this set of Gaussian kernels, without including the Neural Head Basis module to learn moment-specific features. In this way, the Gaussian kernels in canonical space are solely optimized to accommodate multiple lifestages, approaching a mean representation. This approach offers the benefit of enabling the Gaussian kernels to promptly learn the mean head of a person in long-horizon time periods, thereby significantly expediting subsequent training convergence. Following the warm-up phase, we commence optimizing the neural head base while simultaneously fine-tuning the Gaussian Surfels to enhance the mean head representation. This optimization process ensures that both the neural head base and the Gaussian Surfels continually improve and refine the overall representation. The warmup phase lasts for 5000 iterations before the formal training, with applying all loss items except deform regulation.

Table 3: **Hyperparameters during training process.**

| Type | Parameter | Value |
|---|---|---|
| Hashgrid Müller et al. (2022) | Number of levels | 16 |
| | Hash table size | $2^{18}$ |
| | Number of features per entry | 8 |
| | Coarsest resolution | 16 |
| | Finest resolution | 2048 |
| Dynamo(Sec. 3.2.2) | Perset threshold $\kappa$ | 0.0001 |
| | Start iterations $Q$ | 10000 |
| | Iteration interval $q$ | 10000 |
| Weight of loss(Eq. 6) | RGB loss(mouth&eye region) $\lambda_{\mathrm{r}}$ | 40.0 |
| | RGB loss(otherwise) $\lambda_{\mathrm{r}}$ | 1.0 |
| | SSIM loss $\lambda_{\mathrm{s}}$ | 1.0 |
| | LPIPS loss $\lambda_{\mathrm{l}}$ | 1.0 |
| | Depth loss $\lambda_{\mathrm{d}}$ | 1.25 |
| | Normal loss $\lambda_{\mathrm{n}}$ | 1.0 |
| | Consistency loss $\lambda_{\mathrm{c}}$ | 1.0 |
| | Regulation $\lambda_{\mathrm{reg}}$ | 0.01 |

### A.4 TIMEWALKER-1.0

When modeling a lifelong head avatar, existing open-source datasets are limited by a lack of life-stage variations and insufficient data scale, please refer to A.4.3 for the details. To fill this data requirement, we construct a large-scale and high-resolution head dataset of the same individual at different lifestages. In the following section, we will first introduce the construction pipeline of our TimeWalker-1.0 in A.4.1. Then we demonstrate the comprehensiveness of it in lifelong head modeling through **Statistics Analysis** ( A.4.2) and **Comparison with Other Datasets** ( A.4.3).

### A.4.1 TIMEWALKER-1.0 CONSTRUCTION

The data collection process involves the following steps. Initially, we query high-quality videos from YouTube with resolutions greater than 1080P using predefined search prompts to gather a variety of person-specific videos that exhibit diversity in content and appearance. Additionally, for movie stars, we collect their movie sources as supplementary data. Subsequently, an automated video pre-processing pipeline is developed to extract headshots. This involves the detection and cropping

Table 4: *Dataset comparison.* We compare TimeWawlker-1.0 with face datasets from "Lifestage", "Identity", "Age", "Expression", "Frame Count", "Ethnicity", and "Accessory".

| Dataset | Realism | | | | | Diversity | | |
|---|---|---|---|---|---|---|---|---|
| | Resolution | Lifestage | ID | Age | Expression | Frame Count | Ethnicity | Accessory |
| *FFHQ* Karras et al. (2019) | 1024×1024 | ✗ | - | - | ✗ | 70k | ✓ | ✓ |
| *CACD* Chen et al. (2014) | <512×512 | ✓ | 2k | 16-62 | ✓ | 163k | - | - |
| *HRIP* Liang et al. (2016) | 414×464 | ✓ | 4 | - | ✓ | 4k | ✗ | ✗ |
| *INSTA* Zielonka et al. (2023) | 512×512 | ✗ | 12 | - | ✓ | <54k | ✗ | ✗ |
| *RenderMe-360* Pan et al. (2024) | 2448×2048 | ✗ | 500 | 8-80 | ✓ | >243M | ✓ | ✓ |
| *TimeWalker-1.0* | 1024×1024 | ✓ | 40 | 20-80 | ✓ | 2.7M | ✓ | ✓ |

of human faces from the raw videos, achieved by leveraging a pretrained face recognition model to isolate frames featuring the target individual. Following this step, aesthetic assessment models such as HPSv2 Wu et al. (2023b) and LIQE Zhang et al. (2023b) are employed to sift out low-quality head photos from the pool of selected images. Ultimately, the filtered output undergoes a manual review by human evaluators to ensure that the retained headshots meet the requisite quality standards.

### A.4.2 STATISTICS

TimeWalker-1.0 consists of 40 celebrities' lifelong photo collections, with each celebrity containing diverse variations over different lifestages(e.g., shape, headpose, expression, and appearance). The data volume ranges from $15K$ to $260K$ for each celebrity.

We delve into an in-depth exploration of the data distribution within the dataset. *(1) Attributes:* The statistical analysis of the overarching human-centric attributes is delineated in Figure 7, showcasing a broad spectrum of attribute distribution following a long-tail pattern. *(2) Brightness:* The brightness is calculated by averaging the pixels and then converting them to "perceived brightness" Bezryadin et al. (2007). The lower variance of brightness indicates a more similar luminance within the video clip. We analyze the brightness variance from two perspectives: the video level and the celebrity level, where the video level calculates the inter-video brightness variance of the whole dataset while the celebrity level assesses the brightness variance of all the videos belonging to the same celebrity. As depicted in Fig. 8, the collected dataset shows relatively flat changes over inter-video brightness variance, and possesses more diverse lighting changes in each life-long ID. While comparing the inter-video brightness variance with the conventional testbed INSTA dataset for single appearance reconstruction, our dataset embraces more challenges with larger diversity and wider spectrum. *(3) Age, Gender, and Headpose:* As shown in Fig. 9, we demonstrate the age distribution of the TimeWalker-1.0, which indicates our dataset has a balanced age distribution without being biased towards certain age group. Moreover, the dataset includes celebrities of multiple ethnicities (Brown, Yellow, White, and Black). The headpose distribution graph in Figure 9 underscores the diverse range of captured head poses, ideal for the reconstruction of 3D head avatar.

### A.4.3 COMPARISON WITH OTHER DATASETS

We also compare our TimeWalker-1.0 with other datasets to show its superiority. As shown in Tab. 4. The previous dataset either concentrates on 2D head photo generation without 3D supervision, such as FFHQ Karras et al. (2019), or records videos of a specific life-stage where the shape varies are not disputed, such as RenderMe-360 Pan et al. (2024) and INSTA Zielonka et al. (2023). In addition, early datasets like CACD Chen et al. (2014) and HRIP Liang et al. (2016) collect cross-age celebrity data for face recognition and rough head shape modeling. Their data scale and image resolution severely limit them from being implemented for life-long head avatars. Therefore, our TimeWalker-1.0 dataset, stands out in life-stage avatar modeling for its high image resolution, large-scale, wide age range, diverse ethnicity, and most importantly across life-stage data groups.

### A.5 EXPERIMENTS

### A.5.1 PERSONALIZED SPACE VISUALIZATION

**Lifestage**. Fig. 10 demonstrates several individuals with their different lifestages. In part $(a)$ it shows the model's ability to walk through long age period of a person without losing rendering realism, and to represent multiple appearances with diverse skin color. We own this to our powerful Neural Head Basis module which is capable of learning intrinsic features as well as appearance deformation. The disentangled design of neural deformation field and motion warping field enables changing the lifestage but keeping other dimensions like shape, pose and expression unchanged. In part $(b)$, we

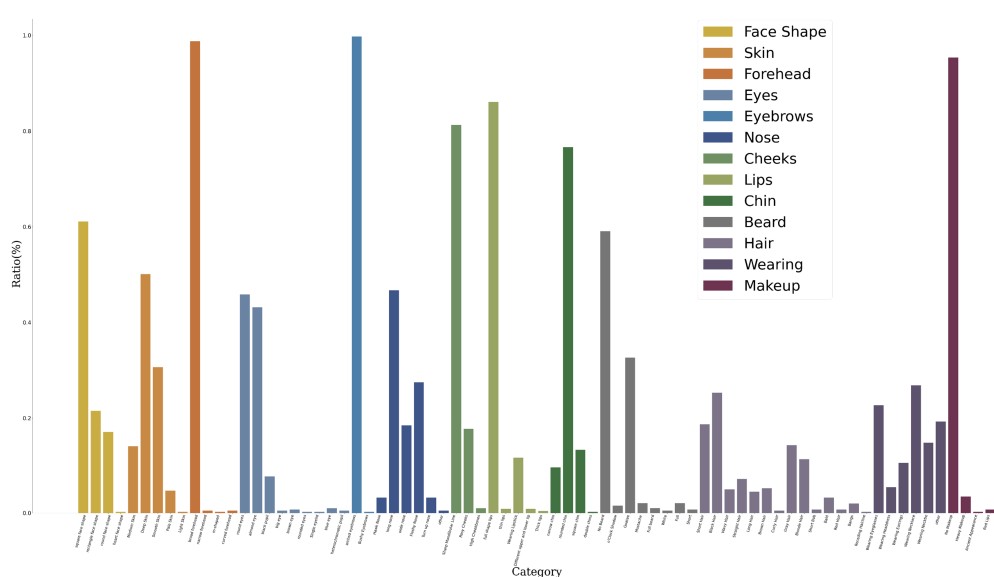

Figure 7: **Statistics: Attributes.** The attribute statistics of TimeWalker-1.0 reveal a wide spectrum of attribute distribution characterized by a long-tail pattern.

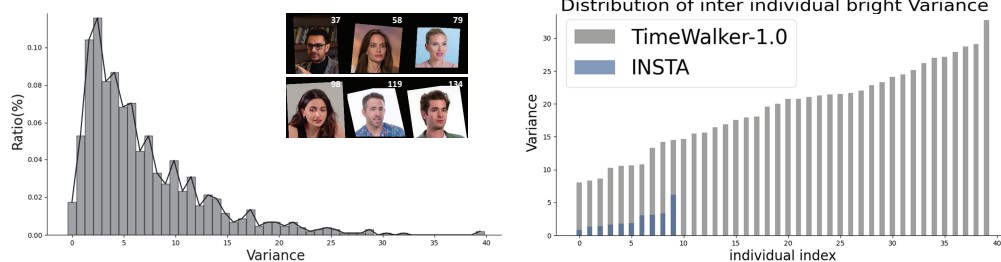

Figure 8: **Statistics: Brightness Variance.** The left part represents the inter-video brightness variance of the dataset, while the right part shows the luminance condition across the whole videos of the same celebrity. The right part illustrates that our dataset enjoys more diverse lighting changes than the INSTA Zielonka et al. (2023) dataset.

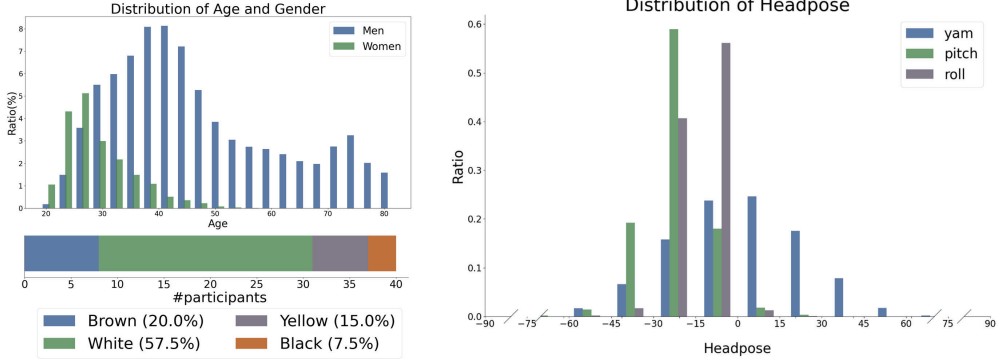

Figure 9: **Statistics: Age, Ethnicity, and Headpose.** The selected videos showcase a diverse range of ages, ethnicities, and headposes. We divide the headpose into 12 clusters with each covering an angle range of $15°$ and we calculate the ratio of each cluster to the total number of headposes as shown in the right figure.

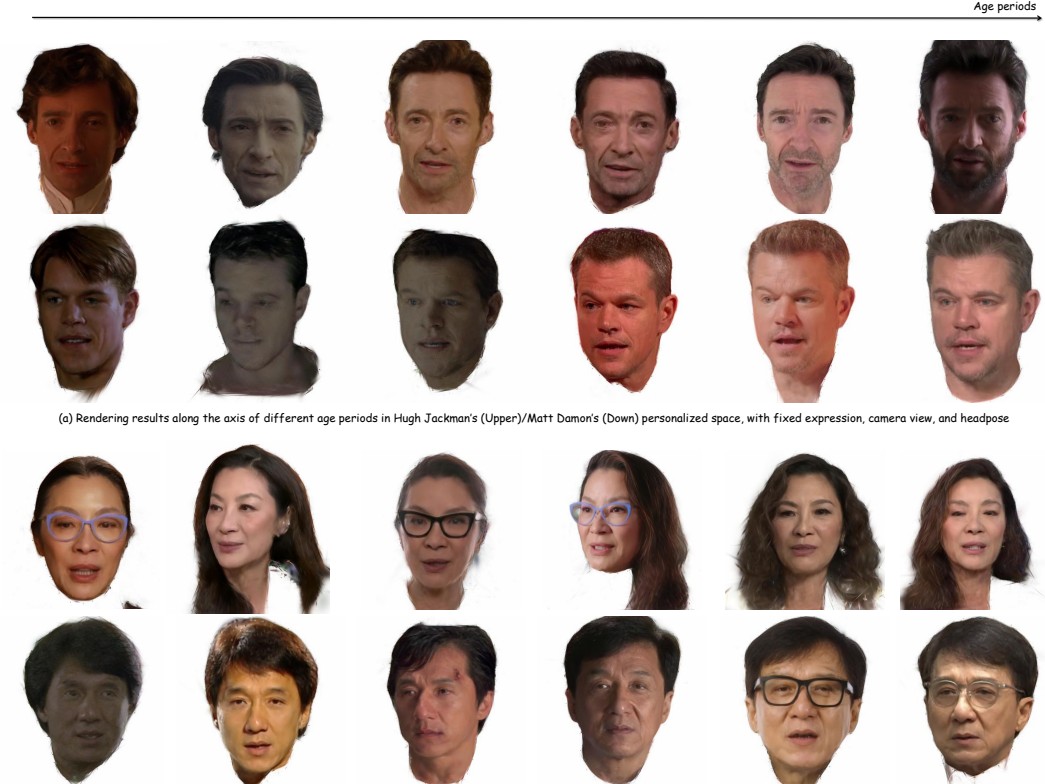

(a) Rendering results along the axis of different age periods in Hugh Jackman's (Upper)/Matt Damon's (Down) personalized space, with fixed expression, camera view, and headpose

(b) Rendering results of Michelle Yeoh (Upper)/ Jackie Chan (Down) under different age periods

Figure 10: **Personalized Space: Lifestage.** We demonstrate multiple individuals and their replicas in different lifestages. (a) We adjust the value in Dynamo to animate the lifestages of individuals, but keep other animation values unchanged. (b) We show the lifestages of more individuals with another ethnicity.

showcase the model's capacity to delineate lifestages within a personalized space across different ethnicities and genders.

### A.5.2 COMPARISONS WITH STATE-OF-THE-ARTS

**Baseline.** We compare our method with state-of-the-art animatable head avatar generation methods and also neural rendering methods, extending from Tab. 1. **INSTA** Zielonka et al. (2023) stands out for its ability to generate high-quality head avatars rapidly, leveraging a multi-resolution hashgrid defined in canonical space to store learned features. For ray points in deformed space, INSTA identifies the nearest triangles of the FLAME tracked mesh and computes the transformation matrix between the tracked mesh and the template mesh. Subsequently, the points are warped back to canonical space, enabling the animation of the avatar using FLAME expression parameters by updating the tracked mesh. To the best of our knowledge, existing state-of-the-art methods for animatable head avatars do not focus on animating the head across diverse time periods. Therefore, in our experiments, we extend the INSTA method to **INSTA++**, enabling it to capture multiple lifestages within a single model. Concretely, we introduce a per-lifestage learnable latent code as a supplementary condition to the density MLP, enabling the storage of lifestage-specific latent information. **FlashAvatar** Xiang et al. (2024) integrates the Gaussian representation with a 3D parametric model by initializing Gaussian points on a 2D UV texture map attached to the mesh surface. It employs a small MLP network conditioned on the Gaussian canonical position and expression code to learn the Gaussian offset, thereby enabling motion driven by facial expressions. **Gaussian-Surfels** Dai et al. (2024) aims to solve the inherent normal and depth ambiguity of 3DGS by cutting one dimension of gaussian ellipsoid, and can reconstruct the dense mesh without losing the realism of static scene rendering. After training, a Poisson mesh with post-process is applied to extract the dense mesh. Notably, although Gaussian Surfels was not specifically designed for animatable head avatars, we include it in our comparison since TimeWalker has adapted the Gaussian Surfels representation for mesh reconstruction, making it a relevant baseline for our evaluation. We further extend the Gaussian-

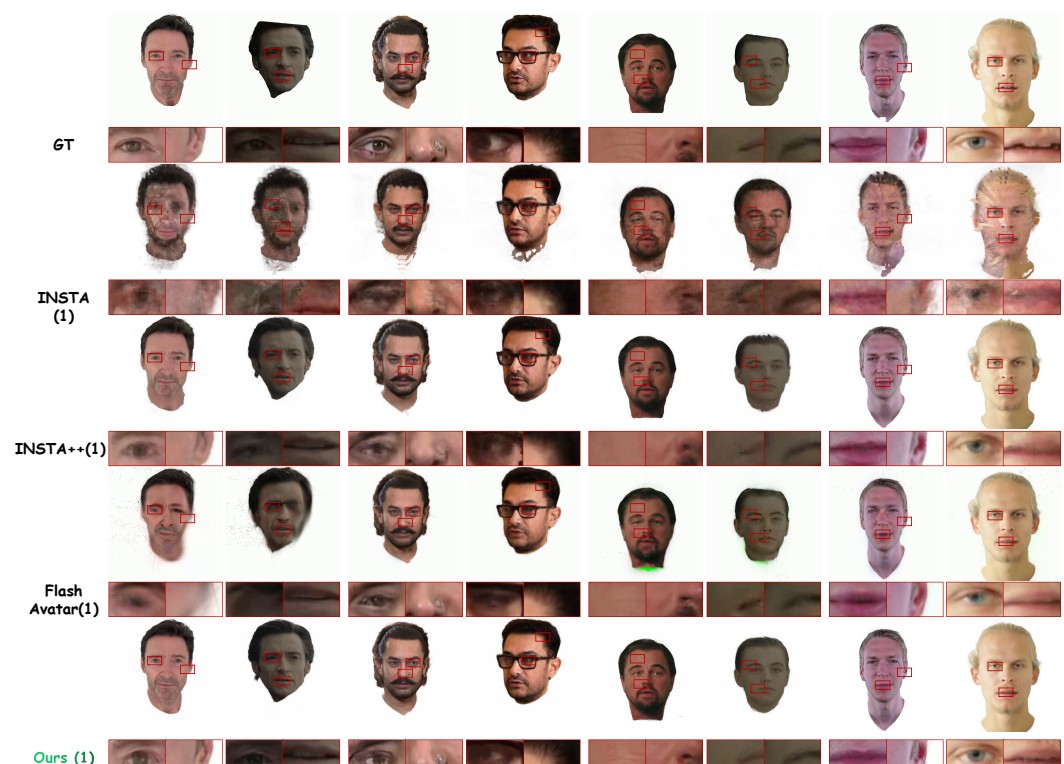

Figure 11: **Qualitative comparison with SOTA with *#Protocol-1***(1 vs 1). All methods, including both ours and the baselines, involve training a single model for each individual, encompassing various lifestages.

surfels to **Gaussian-Surfels++**, a dynamic version to handle motion driven by expression changes. Specifically, we heuristically introduce an MLP-based warping field with expression parameters as conditions, to learn the non-rigid motion in terms of Gaussian attribute shifts.

**Evaluation Protocol.**   We select five representative identities from our TimeWalker-1.0 dataset, each comprising 8-13 lifestages with 8000-20000 frames in total(500-3000 frames for each lifestage). To ensure a balanced evaluation, we address the uneven frame distribution among appearances by allocating the last 10% of frames (capped at 150 frames) from each appearance as the test set and the remaining frames as the training set. This results in evaluating each identity with approximately 800-1300 frames. To further validate the robustness of our method, we conduct an additional experiment on the open-source INSTA dataset, which features 9 subjects with diverse appearances. In our setup, we treat different individuals as distinct appearances of the same person, thereby covering all 9 subjects with a single model. We conduct experiments with two protocals – In 1) *#Protocol-1 (1 vs 1 Comparison)* We train a separate model for each identity, encompassing multiple lifestages. This protocol evaluates the performance of INSTA, INSTA++, and FlashAvatar. In 2) *#Protocol-2 (1 vs N Comparison)* Our model maintains the same pipeline setup as *#Protocol-1*, but for the baseline models (INSTA, FlashAvatar, Gaussian Surfels, and Gaussian Surfels++), we train one model for each lifestage, allowing multiple models for one identity. This allows us to compare the results of our pipeline using a single model against the baseline using multiple models. For evaluation metrics, we utilize three paired-image metrics to assess the quality of individual generated frames: Peak Signal-to-Noise Ratio (PSNR), Structural Similarity (SSIM) and Learned Perceptual Image Patch Similarity (LPIPS). The mean value of each metric is calculated across the test set.

**Results.**   Quantitatively, as shown in Tab. 5 and Tab. 6, our method achieves the best results in the *#Protocol-1*, and best or second best results in the *#Protocol-2* on both our TimeWalker-1.0 dataset and opensource INSTA dataset. Based on the results presented in Tab. 5, our method significantly outperforms other baselines in both protocols when assessed using TimeWalker-1.0. Although INSTA++ shows considerable enhancements post multi-lifestage adaptation, there remains a noticeable gap compared to our approach, underscoring the necessity of meticulously designing our pipeline for creating high-fidelity lifestage replicas. In the 1 vs N Comparison, our results remain

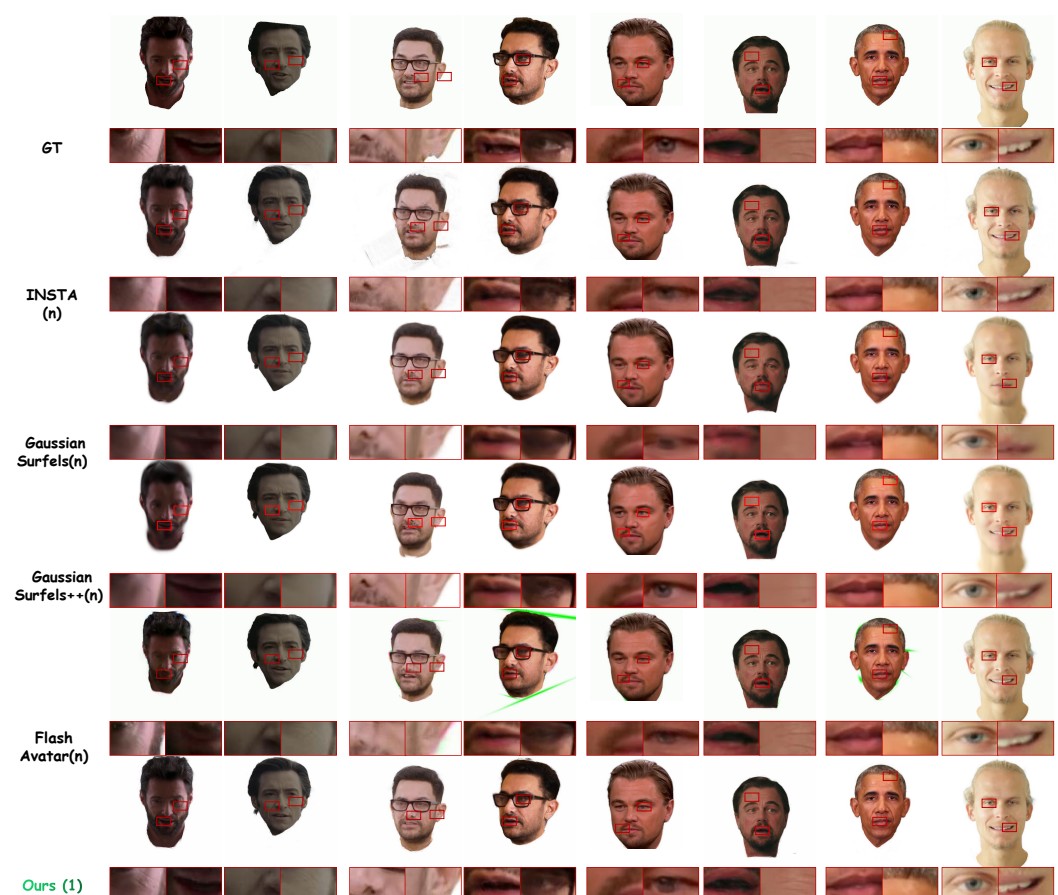

Figure 12: **Qualitative comparison with SOTA with _#Protocol-2_**(1 vs N). In this setting, our method trains one model for one individual across multiple lifestages, while the baselines train multiple models for one individual, _i.e.,_ one model for each lifestage.

competitive, with lower LPIPS values. Similar trends are observed in Tab. 6, where our method surpasses all baselines in #Protocol-1 and delivers competitive outcomes in #Protocol-2. Notably, most pipelines exhibit superior performance in the publicly available INSTA dataset, but they exhibit lower performance in our TimeWalker-1.0 dataset. This disparity can be attributed to the distinct characteristics of the two datasets. The INSTA dataset is curated in a controlled lab environment with consistent lighting conditions, pre-defined head motions, and expressions. In contrast, the data in TimeWalker-1.0 is gathered from real-world scenarios, introducing greater variability in lighting, appearance, accessories, etc. This increased diversity poses significant challenges for constructing personalized spaces.

In Fig. 11, it can be observed that methods like INSTA Zielonka et al. (2023) and FlashAvatar Xiang et al. (2024) struggle to handle appearance variations, resulting in disruptions and blurry outcomes. Even the multi-lifestage extended INSTA++ produces unsatisfactory results with artifacts noticeable in high-frequency areas such as the eyes. Conversely, our method produces rendering outcomes with reduced blurriness and artifacts, showcasing the effectiveness of the pipeline. In #Protocol-2, as indicated Fig. 12, our method showcases the ability to adapt to varying appearances despite being trained with unstructured data. This adaptability enables our method to effectively manage appearance changes of different shapes, yielding competitive outcomes comparable to models trained on singular appearances.

We further compare our method with GANAvatar Kabadayi et al. (2023), a generative head avatar model, to reveal whether the current generative model paradigm is capable of modeling personalized space in the lifelong scale. GANAvatar Kabadayi et al. (2023) utilizes a two-stage training scheme for 3D head avatar, where it firstly leverages a 3D-aware generative model for personalized appearance reconstruction by training on corresponding 2D images. Then a mapping module driven by 3DMM

facial expression parameters is employed for achieving facial expression control on the personalized generative model. As shown in Tab. 7, our method achieves superior quantitative results compared to GANAvatar Kabadayi et al. (2023) on both #Protocol-1 and #Protocol-2, while demonstrating a similar performance trend on the INSTA and TimeWalker-1.0 datasets as the aforementioned baselines. The visual comparisons presented in Fig. 14 indicate that, in the 1 vs N Comparison, GANAvatar struggles with illumination and texture details (e.g., the eyebrows). In the 1 vs 1 Comparison, GANAvatar performs even worse, as it is a personalized generative model that can not effectively manage life-stage appearance variations. As illustrated in block (a) of Fig. 15, the model trained on celebrity life-stage data fails to generate a consistent identity as the 1 vs N Comparison. Additionally, we find that the expression mapping module tends to overfit to specific appearances, as depicted in Fig. 15 block (b). Therefore, single appearance-based methods, like GANAvatar, are not suitable for life-stage head avatar modeling.

Table 5: **Quantitative Evaluation on TimeWalker-1.0.** We evaluate our method with two different protocols. The **Upper** table demonstrates *#Protocol-1* (1 vs 1 Comparison), while the **Lower** table shows *#Protocol-2* (1 vs N Comparison). Pink indicates the best and orange indicates the second.

| Protocol | Method | PSNR↑ | SSIM↑ | LPIPS↓ |
|---|---|---|---|---|
| **1 vs 1 Comparison** | *INSTA* Zielonka et al. (2023) | 20.68 | 0.697 | 0.299 |
| | *INSTA++* | 26.39 | 0.879 | 0.139 |
| | *Flash Avatar* Xiang et al. (2024) | 22.14 | 0.771 | 0.267 |
| | *Ours* | 27.28 | 0.949 | 0.071 |
| **1 vs N Comparison** | *Gaussian Surfels* Dai et al. (2024) | 26.98 | 0.950 | 0.141 |
| | *Gaussian Surfels++* | 27.61 | 0.948 | 0.134 |
| | *INSTA* Zielonka et al. (2023) | 25.47 | 0.86 | 0.170 |
| | *Flash Avatar* Xiang et al. (2024) | 24.9 | 0.848 | 0.165 |
| | *Ours* | 27.28 | 0.949 | 0.071 |

Table 6: **Quantitative Evaluation on INSTA Data.** We evaluate our method with two different protocols. The **Upper** table demonstrates *#Protocol-1* (1 vs 1 Comparison), while the **Lower** table shows *#Protocol-2* (1 vs N Comparison). Pink indicates the best and orange indicates the second.

| Protocol | Method | PSNR↑ | SSIM↑ | LPIPS↓ |
|---|---|---|---|---|
| **1 vs 1 Comparison** | *INSTA* Zielonka et al. (2023) | 19.52 | 0.682 | 0.308 |
| | *INSTA++* | 27.74 | 0.915 | 0.09 |
| | *Flash Avatar* Xiang et al. (2024) | 26.98 | 0.925 | 0.097 |
| | *Ours* | 28.57 | 0.966 | 0.056 |
| **1 vs N Comparison** | *Gaussian Surfels* Dai et al. (2024) | 27.32 | 0.969 | 0.121 |
| | *Gaussian Surfels++* | 27.96 | 0.965 | 0.123 |
| | *INSTA* Zielonka et al. (2023) | 26.98 | 0.935 | 0.077 |
| | *Flash Avatar* Xiang et al. (2024) | 28.15 | 0.925 | 0.096 |
| | *Ours* | 28.57 | 0.966 | 0.056 |

Table 7: **Quantitative Evaluation with Generative Method.** We evaluate our method with a generative method, GANAvatar Kabadayi et al. (2023). Pink indicates the best.

| Method | TimeWalker-1.0 | | | INSTA Data | | |
|---|---|---|---|---|---|---|
| | PSNR↑ | SSIM↑ | LPIPS↓ | PSNR↑ | SSIM↑ | LPIPS↓ |
| **GANAvatar(1 vs 1)** | 15.40 | 0.809 | 0.227 | 18.15 | 0.832 | 0.172 |
| **GANAvatar(1 vs N)** | 18.44 | 0.852 | 0.151 | 24.76 | 0.900 | 0.0637 |
| **Ours** | 27.28 | 0.949 | 0.071 | 28.57 | 0.966 | 0.056 |

### A.5.3 MESH COMPARISON

The effectiveness of our mesh reconstruction pipeline is evaluated through a comparison of meshes extracted from various pipelines: Colmap Schönberger & Frahm (2016), INSTA++, Gaussian Surfels Dai et al. (2024), and our own methodology. Data from a single lifestage is provided to Colmap and Gaussian Surfels, tailored for static scene reconstruction. Conversely, our pipeline and INSTA++, designed for handling multiple lifestages, receive all data of an individual. As demonstrated in Fig. 13, our pipeline successfully reconstructs the head mesh with precise shape,

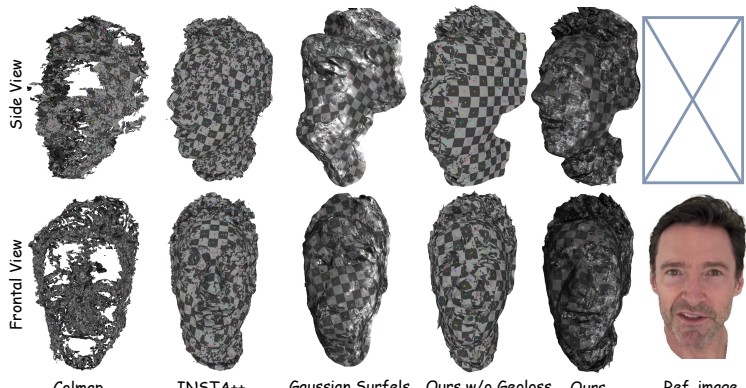

Figure 13: **Static Mesh Comparison.** We visualize and compare static mesh reconstruction results from Gaussian Surfels Dai et al. (2024) and Ours (adaptive version of GS with DNA-2DGS). We render the meshes in both frontal and side views using Blender software under identical rendering conditions. Despite these consistent settings, rendering results exhibit significant differences due to the diverse topologies of the meshes.

while the original Gaussian Surfel and Colmap fail to derive a meaningful mesh. This disparity is partly attributed to the restricted data from a single appearance and camera pose. Both pipelines struggle to recreate a plausible head mesh without prior knowledge of human head anatomy, given that monocular reconstruction is inherently an ill-posed problem. In contrast, our mesh reconstruction, utilizing data from multiple lifestages to encompass a broader spectrum of head poses and camera angles, achieves superior quality with more accurate surface details. Although INSTA++ can also reconstruct a plausible head mesh, it exhibits a higher level of noise. This comparison underscores the efficacy of our DNA-2DGS component in reconstructing head meshes from unstructured photo collections.

**Time cost in meshing process.** In the context of single mesh generation, our approach necessitates approximately 10 minutes, a timeframe comparable to or longer than that required by Gaussian Surfels and INSTA++. Notably, our methodology exhibits a notable advantage in the creation of mesh sequences, as we leverage direct animation of the static mesh to generate a new mesh. In contrast, Gaussian Surfels and INSTA++ are compelled to iteratively execute the entire mesh generation process, leading to a linear growth in time consumption. Colmap's inability to generate mesh sequences stems from its requirement of the complete sequence to reconstruct a single mesh.

### A.5.4 MORE ID RESULTS

To show the generalization and effectiveness of our pipeline, we demonstrate more individual results from different genders and ethnicities in Fig. 16.

### A.6 APPLICATION

**3D Editing.** To confirm the validity of our neural parametric model that produces 3D consistent and personalized outcomes, we conduct additional experiments in the 3D head editing application. We evaluate the results of our Gaussian Surfels-based approach by referring to DGE Chen et al. (2024), which performs direct 3DGS editing with a pre-trained diffusion model while maintaining multi-view consistency. During the optimization, we switch the representation to Gaussian Surfels and keep the other settings unchanged from the original implementation. Fig. 17 presents a visual comparison of the re-rendered results, based on various editing prompts, highlighting the 3D consistency achieved by our method. The direct editing method fails to disentangle animated expressions (as can be observed from the visual result of the text prompt: "Add smile"), and it also undergoes other unforeseen changes, such as alterations in skin color. In contrast, our method executes disentangled and fine-grained multi-dimensional animation. Focusing on the final column of the illustration, the editing performance of the original implementation (DGE with 3DGS) within our TimeWalker-1.0 dataset exhibits shortcomings, primarily attributed to the following factors. Primarily, noticeable blurring is evident in both the original rendering and the edited output, particularly pronounced in the facial features such as the eyes and mouth. Given that 3DGS is designed to handle static objects rather than dynamic data sequences, areas that undergo frequent modifications lead to inconsistencies and

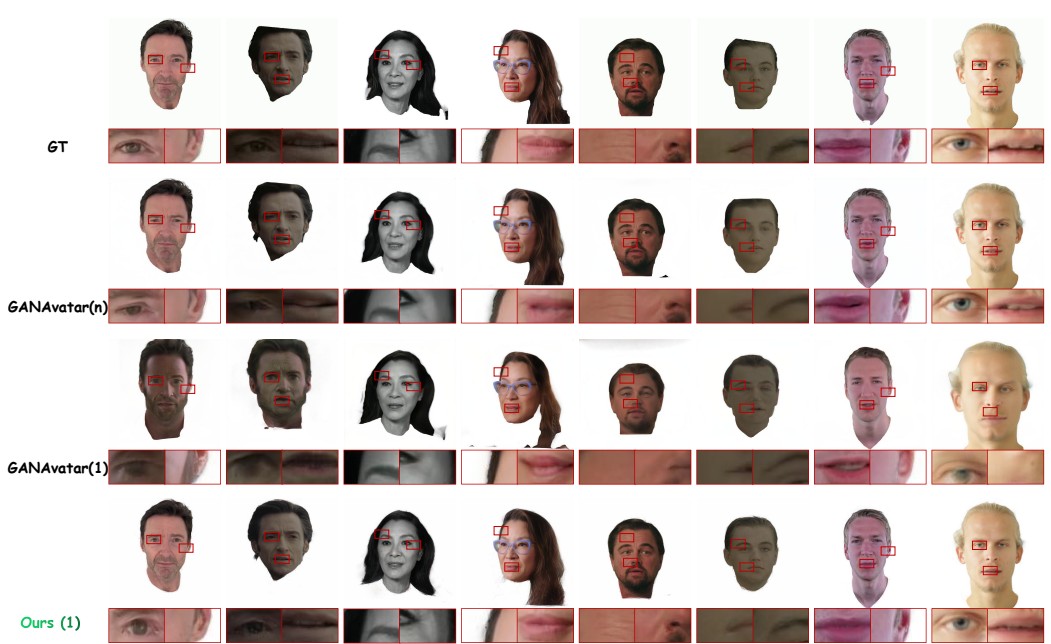

Figure 14: **Qualitative Comparison with Generative Method.** Symbol $(1)$ in figure means train one model for each individual, symbol $(n)$ means train $N$ models for each individual, *i.e.,* one model for each lifestage.

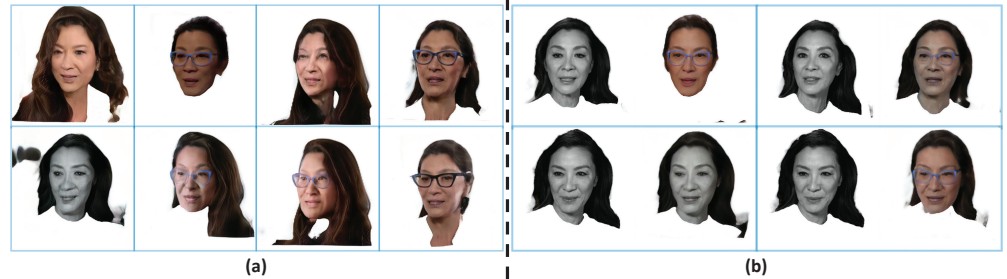

Figure 15: **Sampled Results of GANAvatar.** In the #Protocal-1, (a) GANAvatar struggles to generate consistent identity, which is rooted in (b) the expression mapping network overfits on different lifestages. Even close expressions could generate quite different head images.

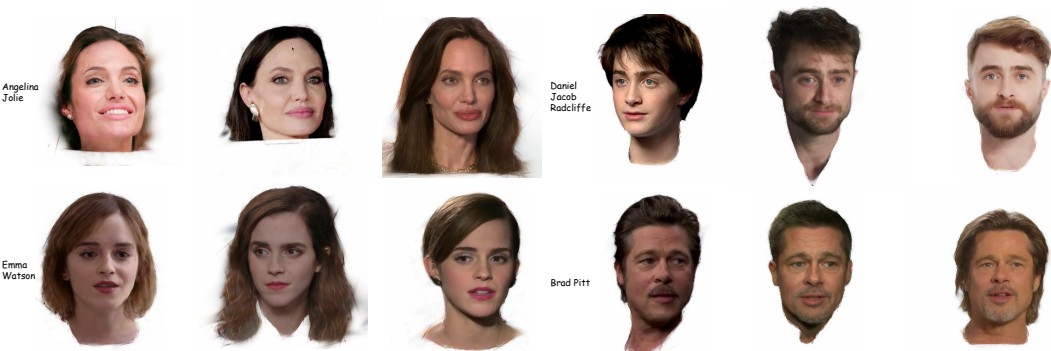

Figure 16: **More sampled ID result.** We visualize more individuals with different genders.

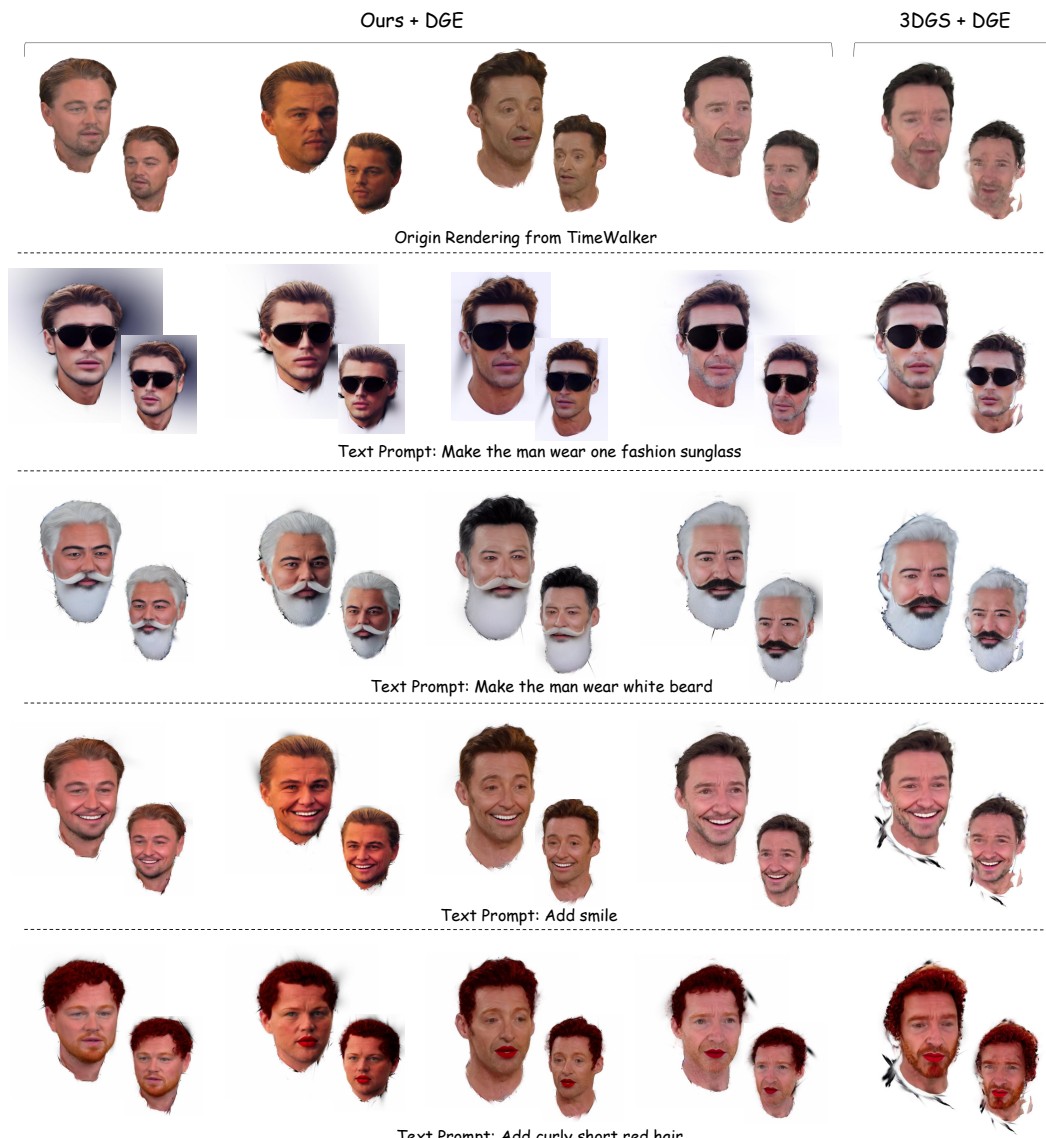

Figure 17: **Visualization of Downstream Application on 3D Editing with DGE.** We present the editing outcomes elicited by a variety of text prompts. The upper two examples illustrate the capability of DGE Chen et al. (2024) method to introduce new elements to the human head model, while the lower two examples illustrate how it functions when altering the attributes of head components. We compare the editing results between our models with DGE, and the original 3DGS with DGE.

produce blurred outcomes. Furthermore, the inadequate diversity in camera pose distribution within the training data at a single lifestage results in depth ambiguity, notably observable in the neck region. This issue persists after 3D editing, manifesting as discontinuities and dark artifacts encircling the neck. In contrast, our model effectively addresses this challenge through the automated interpolation of data from various lifestages during training, resulting in a more coherent geometric representation.

## A.7 BROADER IMPACT

In this study, our objective is to accurately generate images of an individual, focusing solely on rendering head avatars and seamlessly altering their appearance within a predefined set of lifestages. It is noteworthy that our work does not endeavor to create fictitious motions or animations; rather, we strive to faithfully represent the subject's appearance and visible views. Due to its sensitivity, we strictly obey the non-commercial license for the dataset and will take more necessary items before it is released.

