# OpenReview forum: "TimeWalker: Personalized Neural Space for Life-long Head Avatar"
_ICLR.cc/2025/Conference — ICLR 2025 Conference Withdrawn Submission_

### Official Review · Reviewer_Btrk · 2024-10-17

**Soundness:** 4
**Presentation:** 3
**Contribution:** 3
**Rating:** 6
**Confidence:** 4

**Summary:**

The paper presents a novel framework to model lifelong 3D animatable avatars. To address this avatar reconstruction problem on a lifelong scale, the author first constructs a large-scale dataset comprising lifelong images from different subjects. Then, they propose a head representation using neural head basis and 2D Gaussian Splatting (GS) for modeling all life stages of one person. Extensive experiments demonstrate the proposed approach can effectively reconstruct lifelong 3D animatable avatars.

**Strengths:**

1. The paper addresses an intriguing and technically significant task, and the constructed dataset will be valuable for future research in this domain.

2. The proposed method is technically robust, and the extensive experiments provide strong justification for its effectiveness.

**Weaknesses:**

1. The reconstructed lifelong avatars exhibit significant artifacts beyond those associated with the mouth region, as noted by the authors. However, there is a lack of discussion regarding the origins of these artifacts. For instance, could the limited availability of data during certain life stages be a contributing factor?

2. It would be beneficial to explore the advantages of incorporating multiple lifetimes within the representation for this task. However, the paper lacks a discussion on this point, particularly concerning the extent of performance improvement attributed to the inclusion of additional lifelong data.

3. The novelty of the proposed method appears to be somewhat constrained. The use of blendshape concepts within the Gaussian Splatting framework bears similarities to prior work [1]. Additionally, the adoption of the multi-resolution Hashgrid has also been utilized in INSTA [2], further limiting the originality of the approach.


[1] Ma, Shengjie, et al. "3d gaussian blendshapes for head avatar animation." *ACM SIGGRAPH 2024 Conference Papers*. 2024.

[2] Zielonka, Wojciech, Timo Bolkart, and Justus Thies. "Instant volumetric head avatars." Proceedings of the IEEE/CVF Conference on Computer Vision and Pattern Recognition. 2023.

**Questions:**

1. Regarding the FLAME parameters of the dataset:
   1. What method was employed to obtain the FLAME parameters for the constructed dataset?
   2. How accurate are these FLAME parameters? Additionally, how might the accuracy of the preprocessed FLAME parameters impact the fidelity of the final avatars?

2. Is the representation of lifetime spaces continuous?

3. What is the performance of the method from different viewpoints? The results presented primarily focus on frontal views.

4. The specific lifetime avatars exhibit noticeable shadowing. Given the videos captured at different times, could it be possible to mitigate these shadows to enhance the visual quality of the results?

---

> ### Author Response · Authors · 2024-11-28
> **Technical response - Part 1**
>
> >**Q1**: The reconstructed lifelong avatars exhibit significant artifacts beyond those associated with the mouth region, as noted by the authors. However, there is a lack of discussion regarding the origins of these artifacts. For instance, could the limited availability of data during certain life stages be a contributing factor?
>
> **A1**: Due to the page limitations of the main paper, we briefly address this in Section 5. And yes, data limitations contribute to the observed artifacts, as the mouth region is often occluded, leading to incomplete supervision during training. This results in artifacts during exaggerated expressions, as shown in Figure 6. Incorporating generative priors or curated datasets specifically for the mouth region could help mitigate this limitation.
>
>
> >**Q2**: It would be beneficial to explore the advantages of incorporating multiple lifetimes within the representation for this task. However, the paper lacks a discussion on this point, particularly concerning the extent of performance improvement attributed to the inclusion of additional lifelong data.
>
> **A2**: We discussed the benefits in Sec.3.5 Lines 424-430 and will further emphasize these in the revised version. In summary,  our model  learns a person's personalized space with the support of neural "mean head" and principal head basis, making the space steerable and fostering high-quality results under various settings.
>
> Key advantages include:  **full-scale animation,  reenactment, high-quality rendering,  disentanglement of attributes, identity preservation, and enhancement of downstream task performance**. Additionally, incorporating multiple lifetimes within the representation **helps compensate for data scarcity**, particularly in geometry reconstruction and producing side-view rendering outcomes. Refer to the answer to your Q7 for more details on this benefit.
>
>
> >**Q3**: The use of blendshape concepts within the Gaussian Splatting framework bears similarities to prior work. "3d gaussian blendshapes for head avatar animation.(3DGS-B)"
>
> **A3**: The concepts differ fundamentally between our work and 3DGS-B in both **modeling scope** and **complexity**. 3DGS-B focuses on and can only learn a linear expression space under a single appearance modeling condition. In contrast, our framework, tackles the more complex challenge of lifelong modeling, serving as a first baseline to construct a neural personalized space towards life-long scale. This space captures not only expression variations but also shared and distinct features across multiple life stages. It achieves compelling attribute disentanglement, enabling comprehensive modeling and dynamic, multidimensional animations of personalized head avatars under diverse conditions. Besides, additional by-products (as replied in the last questions) can be inherently generated from our  neural personalized space.
>
> One more  important point we would like to highlight  is that the vanilla blendshape concept has been extensively applied in various studies combined with different representations (e.g., Nersemble [2], NerfBlendshape [4]). However, these approaches focus on modeling momentary-level spaces. The novelty focus for no matther  these papers, or our work, is not about simply combining blendshape concepts with Gaussian Splatting or NeRF, but in how the space could be learned and the unique properties of the learned space. Building on such a classic linear combination concept, we have developed a neural personalized space that redefines the scale and adaptability of head avatar modeling.
>
>
>
> >**Q4**: The adoption of the multi-resolution Hashgrid has also been utilized in INSTA, further limiting the originality of the approach.
>
> **A4**: The multi-resolution hashgrid is a foundational representation introduced by instant-ngp [1] and has been widely adopted as a common strategy across various fields. For head modeling, for instance, INSTA utilizes a single multi-resolution hashgrid to directly store head avatar features, while NerSemble [2] employs a fixed number of hashgrids.
>
> In contrast, our approach **dynamically adjusts the number** of multi-resolution hashgrids during training. This adaptive design allows the model to capture deeper characteristics rather than merely memorizing superficial appearances, enabling more comprehensive and personalized head avatar representations (Refer to Sec. 3.2.2, Lines 297–315).

---

> ### Author Response · Authors · 2024-11-28
> **Technical response - Part 2**
>
> >**Q5**: What method was employed to obtain the FLAME parameters for the constructed dataset?How accurate are these FLAME parameters? Additionally, how might the accuracy of the preprocessed FLAME parameters impact the fidelity of the final avatars?
>
> **A5**: We adopted the FLAME fitting preprocessing inspired by INSTA with the following step: (1) MICA[5]-based FLAME parameter initialization; (2) Optimization process based on the Metrical Photometric Tracker. Given the discontinuous nature (camera switching) of the camera within one life stage, this step needs additionally heightened the learning rate for the camera extrinsics. (3) Post-optimized filtering. Owing to the increased noise inherent in in-the-wild data, the accuracy of FLAME parameters tends to be lower compared to data from "standard inputs". To mitigate this, we implemented a filtering rule post-FLAME fitting to eliminate low-quality FLAME parameters.
>
> In general, substandard FLAME parameters can introduce artifacts in rendering results and diminish the performance of expression animation. Refer to Fig. 6 to observe how the less accurate FLAME result impacts the intensity of the animation with exaggerated expressions.
>
>
> >**Q6**: Is the representation of lifetime spaces continuous?
>
> **A6**: Yes, the demo on our project page also demonstrated the continuity of personalized space in multiple scales.
>
> >**Q7**: What is the performance of the method from different viewpoints? The results presented primarily focus on frontal views.
>
> **A7**: The head pose distribution in our Timewalker-1.0 dataset exhibits a predominantly **long-tail distribution**, as illustrated in Fig. 9. This distribution highlights the insufficiency of side view data for modeling identity features at wide angles. Thus, our emphasis is primarily on rendering viewpoints around views with angles inside 30 degree,  to showcase the effectiveness of our method in multi-dimensional reconstructions and animations
>
>  Two observations for sideview: (1)our model **outperforms other SOTAs** even when confronted with missing view data. By constructing a comprehensive personalized space across different lifestages. This can capture the overall representation of the identity. When certain view data is unavailable, our model generates the view renderings that reflect the average representation of the identity, rather than merely introducing artifacts. (2) While effective within **moderate view ranges** (Refer to Fig. 17 Row 1, the visualization of the novel view rendering), the model struggles to handle extreme side views like 90 degree. In the revised version, we will expand the discussion on the performance of side-view rendering.
>
> >**Q8**: The specific lifetime avatars exhibit noticeable shadowing. Given the videos captured at different times, could it be possible to mitigate these shadows to enhance the visual quality of the results?
>
> **A8**: To eliminate shadowing from the rendering output, the model must essentially learn features that are independent of shadowing effects. Several potential approaches can be considered: (1) Incorporate the modeling of light effects by including albedo map prediction during model training; (2) Utilize the robust capabilities of pre-trained generative models to address and mitigate shadowing issues effectively.
>
> **Citation**
> ```
> [1] Müller T, Evans A, Schied C, et al. Instant neural graphics primitives with a multiresolution hash encoding. In ACM  TOG 2022.
> [2] Kirschstein T, Qian S, Giebenhain S, et al. Nersemble: Multi-view radiance field reconstruction of human heads. In ACM  TOG 2023.
> [3] Zheng Y, Yifan W, Wetzstein G, et al. Pointavatar: Deformable point-based head avatars from videos. In CVPR 2023.
> [4] Gao et al Reconstructing Personalized Semantic Facial NeRF Models From Monocular Video. In SIGGRAPH Asia 2022.
> [5] Zielonka W, Bolkart T, Thies J. Towards metrical reconstruction of human faces. In ECCV 2022.
> ```

---

### Official Review · Reviewer_g9vf · 2024-10-23

**Soundness:** 3
**Presentation:** 3
**Contribution:** 3
**Rating:** 3
**Confidence:** 4

**Summary:**

This paper proposes to reconstruct a 3D head avatar of a person on a lifelong scale. Specifically, unlike previous avatars that are limited to moment-level reconstructions, this paper achieves full reconstruction and animation of a person across different ages. To this end, they introduce a new head parameter model, in which they design a set of head basis. By linearly combining these bases, they can model a person’s face at different life moments. Additionally, they propose a DNA-2DGS to better integrate the head basis with the deformation from the original FLAME model, resulting in realistic head deformation modeling. They also introduce a dataset where each ID consists of 12K -260K frames across diverse age and head distributions.

**Strengths:**

1.  They present a new avatar setting that focuses on capturing a comprehensive head avatar over a lifelong scale. To achieve this, they introduce a dataset where each ID contains 12K to 260K frames, spanning diverse age groups and head variations.
2.  This paper proposes a Dynamo module, which learns a set of head basis that could represent the comprehensive head variations of the target person over ages.
3.  They introduce DNA-2DGS to model head motion deformations based both on head basis and FLAME parameters.
4.  Extensive experiments are conducted to verify the effectiveness of the proposed method.

**Weaknesses:**

1.  The specific role of this residual embedding is unclear. Is it because the expressiveness of the head basis is insufficient? There seems to be a lack of ablation studies.
2.  Some details are missing. For example, during inference, how are the blend weights for head basis determined? For instance, when the age is 20, how is the age-to-blend weight mapping obtained? Additionally, is the  I_{res}  that was consistent for all age stages?
3.  The results contain several artifacts, especially during eye-blinking and around the head edges.
4.  In the video results, different life stages exhibit varying lighting conditions. That means several age-irrelevant attributes, such as lighting, were not decoupled from age.

**Similar/identical paper published at CVPR 2024 Workshop POETS 2nd Round**
Thanks for Reviewer f3xJ's remainder.
After reading the paper https://openreview.net/forum?id=3cUdmVfRb4,  I found that this paper is very similar to it, even with identical experimental values (Ablation Study). Based on these findings, I decided to reject this paper.

**Questions:**

1. Why not use a general basis instead of having a specific head basis for each person?
2. How many head bases are learned for each person?
3. What is the inference speed?

---

> ### Author Response · Authors · 2024-11-28
> **Technical response**
>
> >**Q1**: The specific role of this residual embedding is unclear. Is it because the expressiveness of the head basis is insufficient? There seems to be a lack of ablation studies. Is the I_{res} that was consistent for all age stages?
>
> **A1**: The residual embedding I_{res} constitutes learnable parameters that serve as global compensators for each lifestage, like adjusting lighting conditions. Through the incorporation of residual embedding in our pipeline, we partially disentangle lighting effects from lifestage-specific attributes, allowing lighting adjustments while preserving the core characteristics of each lifestage. In our revision, we will include an analysis and discussion to clarify the specific role and impact of the residual embedding.
>
>
> >**Q2**: For example, during inference, how are the blend weights for head basis determined?
>
> **A2**: After model training, for each lifestage, a set of specific blending weights are learned and assigned to the multiple multi-solution hashgrids for each lifestage. By adjusting these blending weight coefficients, we can exercise precise control over the progression of lifestages.
>
> >**Q3**: When the age is 20, how is the age-to-blend weight mapping obtained?
>
> **A3**: Not sure if we understand your question correctly. If you are inquiring about the blending weight associated with the specific age of 20 for an identity, it is important to note that our model does not explicitly define the concept of "Age." Instead, it constructs an interpolatable personalized space based on the notion of *lifestage* (refer to Sec.A.2). However,  as a possible solution, an age-to-blend-weight mapping can be approximated for each identity by labeling specific ages with the corresponding lifestages and their blending weights.
>
> >**Q4**: The results contain several artifacts, especially during eye-blinking and around the head edges.
>
> **A4**: In Sec.5 Lines 527-533, we shortly discuss how data preprocessing inaccuracies and data constraints contribute to rendering artifacts. Here is more concrete unfolding on these two types of artifacts:
>
> (1) **Eye-blinking:** our dataset, curated from in-the-wild unstructured sources, inherently exhibits a long-tail distribution. While our method addresses data imbalance to some degree, rare occurrences of eye-blinking during training can still result in artifacts during these sequences in inference.
>
> (2) **Head edges:** the discrepancies observed around the head edges stem from inaccuracies in the image background matting process. The complex and varied backgrounds in in-the-wild data can result in imprecise matting, introducing additional noise in the region around the head edges during training. These challenges underscore the need in the future to develop methods for creating a high-fidelity head avatar that minimizes its reliance on data preprocessing techniques.
>
> >**Q5**: In the video results, different life stages exhibit varying lighting conditions. That means several age-irrelevant attributes, such as lighting, were not decoupled from age.
>
> **A5**: The diverse lighting conditions present across different lifestages pose significant challenges to fine-grained attribute disentanglement for constructing head avatars. By integrating residual embedding into our pipeline, we can partially separate lighting effects from lifestage attributes. For better disentanglement of lighting effects, some possible solutions could be: (1) incorporating Albedo map modeling into the pipeline to isolate the intrinsic properties of the subject's appearance from external lighting effects; (2) adding lighting augmentations and equivalent representation learning during training,  foster invariant representations robust to lighting variations.
>
> >**Q6**: Why not use a general basis instead of having a specific head basis for each person?
>
> **A6**: There is a trade-off between opting for a general basis vs a person-specific head basis. Employing a general basis such as the 3DMM style enables the generation of a head avatar in a standardized representation, albeit at the cost of losing individual personal characteristics. Conversely, our objective is to create a personalized space capable of accurately and effectively *capturing changes across various lifestages while preserving common characteristics and personality*. If you want to establish a general basis for multiple individuals,  constructing a conditional space with instance-specific designs to compensate for missing details would be a more effective approach.
>
>
> >**Q7**: How many head bases are learned for each person?
>
> **A7**: The number of head bases for each person is dynamically learned by the network, and typically ranges  from 6 to 10 in our experiments. Refer to Sec.3.2.2 Lines 297-315, in which we answer the question "How to set the number of head basis? " in detail.
>
> >**Q8**: What is the inference speed?
>
> **A8**: Refer to reviewer TANo Q5.

---

### Official Review · Reviewer_TANo · 2024-11-03

**Soundness:** 2
**Presentation:** 2
**Contribution:** 1
**Rating:** 3
**Confidence:** 4

**Summary:**

This paper proposes TimeWalker: a scheme for modeling lifelong head portraits. The method learns from a sequence of images and videos of the target object.

** Here is the updated review that ignores the workshop paper and incorporates new content from the rebuttal. **

The rebuttal addressed concerns regarding technical details and comparison baselines, allowing for a better evaluation of the work, but some concerns still remain:
1. The proposed method introduces a novel task, enabling reconstruction on unconstrained noisy data while achieving rendering across time. However, artifacts remain a significant limitation to the method's practical applications. (Supp video 10:40~11:30)
2. While the primary focus of this paper is to establish a neural personalized space spanning various life stages, smooth transitions across stages is an integral component. Abrupt changes will impair the task performance.
3. The method leverages images across lifelong times, one-shot approach can reconstruct avatars by making inferences on these images. This is why I think the task in this paper has some overlap with these approaches: although they do not construct all the avatars in one neural space, the results is somewhat similar (especially the time switching of this method is not smooth, which may be similar to inference on different images).

** The end of the updated part. **

**Strengths:**

- The article can learn head reconstruction from a set of videos and pictures of the target object at different times, and change between the image and shape at different times after learning is completed.
- This paper includes a dataset of 40 celebrities.

**Weaknesses:**

- The visual effects of this method have many artifacts, which can usually be avoided when using a large amount of data (15k-260k per person). In addition, the transitions between different time periods are not natural enough, which has a significant impact on achieving the main goal of this method: seamlessly altering their appearance within a predefined set of life stages. For example, in the Personalized Space provided by the project website, the texture of the skin and the overall blurriness are unsatisfactory, and the lifestage switching looks like a quick and abrupt switch between videos of several life periods. The authors should also consider showing more seamless switching results.
- Most importantly, the approach has not been thoroughly compared, making it difficult to assess its contribution. For example, the method lacks discussion of recent Gaussian head methods[1,2], but compared to the baseline methods (insta) of these papers[1,2],
- There is also a lack of comparison with one-shot head reconstruction methods [3,4,5,6], which can also easily reconstruct the target person's head of various ages and shapes, given that in our setting, we have data of the target person at different ages. Although these one-shot methods cannot achieve seamless switching between different shapes and life cycles, this method is also not fully evaluated in this regard (see questions item 2,3).
- While there is a similar paper (https://openreview.net/forum?id=3cUdmVfRb4) published in non-archived form and allowed to be resubmitted in archived form, the current submission reuses a lot of text/visualizations/quantitative results, importantly the methodological and data details are different while the experimental results are the same. For example, the dataset TimeWalker-1.0 proposed in the non-archived paper includes 20 celebrities, but in this submission there is 40. But the ablation study table: table 2 in this submission reused all the numbers in table1 of the non-archived paper.

- Reference:

        [1] Gaussian Head Avatar: Ultra High-fidelity Head Avatar via Dynamic Gaussians
        [2] SplattingAvatar: Realistic Real-Time Human Avatars with Mesh-Embedded Gaussian Splatting
        [3] Generalizable One-shot Neural Head Avatar
        [4] Real3D-Portrait: One-shot Realistic 3D Talking Portrait Synthesis
        [5] GPAvatar: Generalizable and Precise Head Avatar from Image(s)
        [6] Portrait4D-v2: Pseudo Multi-View Data Creates Better 4D Head Synthesizer

**Questions:**

- How fast is the rendering of this method during inference?
- The method does not evaluate the identity preservation of the person when controlling for lifestage and shape.
- When switching between different life stages, do you consider the order of switching time? Or multiple life stages switched randomly?
- It seems that the Gaussian and 3DMM of this method are bound together (during initialization). Why is it that when it is driven by moment (change the expression), it does not directly use 3DMM to deform the Gaussian points, but introduces a warp field instead?
- In Table 5 (in 1 vs N), FlashAvatar reports lower results than INSTA. The experiments (1vsN) has a similar setting to FlashAvatar paper, but the results is inconsistent with the statement in the FlashAvatar paper that FlashAvatar performs better than INSTA. What could be the possible reason?
- Please refer to the weakness.

**Details Of Ethics Concerns:**

This paper introduces a new dataset including 40 celebrities.

---

> ### Author Response · Authors · 2024-11-28
> **Technical response - Part 1**
>
> >**Q1**: The visual effects of this method have many artifacts, which can usually be avoided when using a large amount of data (15k-260k per person).
>
> **A1**:
>
> (1) To clarify,  we use 8,000–20,000 images per person, with train-test splits detailed in Section A.5.2 (Lines 1059-1060). For reference,  single-appearance head reconstruction  paradigms like INSTA [4], I AM Avatar [2], PointAvatar [3] ,and GaussianAvatars [5] typically use 1,700–5,368 images. Our dataset spans all life stages, making the data allocation per stage relatively modest.
>
> (2)  Larger in-the-wild datasets often introduce noise, imbalance, and challenges rather than improvements. Similar observation has also been shown in other related human-centric  fields (e.g., Li et al, StyleGAN-Human, ECCV 2024). Also, scaling data volume does not directly address entangled issues across dimensions.  Concrete evidence supporting the reviewer's claim is needed.
>
>  (3)  Subjective assessments vary. Our method demonstrates better quantitative and qualitative performance over SOTAs (Sec.5.2, Lines 1074-1144, Tab. 5&6, Fig.11&12). Providing examples with better results than ours would strengthen your review argument.  While not perfect, we aim our work  as a baseline for this new task, and the remaining challenges could also drive future advancements in lifelong personalized avatar learning.
>
> >**Q2**:  The transitions between different time periods are not natural enough, which has a significant impact on achieving the main goal of this method: seamlessly altering their appearance within a predefined set of life stages.
>
> **A2**: Our primary focus in this paper is the establishment of a neural personalized space spanning various life stages, which could benefit reconstruction and animation in multiple dimensions (e.g., shape/appearance/expression/) across lifestages ,  rather than solely transitions between different time periods. (as stated in Sec.3.5 Lines 424-430)
>
>
>
> >**Q3**: Most importantly, the approach has not been thoroughly compared, making it difficult to assess its contribution. For example, the method lacks discussion of recent Gaussian head methods (Gaussian Head Avatar/SplattingAvatar)
>
> **A3**:
> (1) We  conduct a thorough comparison  (Sec.5.2, Lines 1074-1144, Tab. 5&6, Fig.11&12) based on several critical criteria to reveal the factors that might influence the accomplishment of the task: **(a) Base representations.** Thus we  categorize and select representative SOTA methods into NeRF-based (INSTA, and modified version INSTA++), Gaussian Splatting-based (FlashAvatar), and generative methods (GANAvatar). The selection of Flashavatar as a representative state-of-the-art Gaussian head method is deliberate. **(b) Gaussian Splatting variants.** To further explore the impact of different design choices within the base Gaussian Splatting (GS) paradigm, we provide an in-depth evaluation of GS variants (Gaussian Surfels, Gaussian Surfels++). This ensures a precise analysis of GS impacts and limits. It would be better if the reviewer could list the necessity and essence to compare with Gaussian Head Avatar/SplattingAvatar, and the reason to support the "most important" claim of comparing them.
>
> >**Q4**: There is also a lack of comparison with one-shot head reconstruction methods.
>
> **A4**: The references highlighted by the reviewer  predominantly concentrate on:
>
> (1) Generalizable On-shot NHA/Portrait4D-v2/GPAVATAR: head reconstruction and animation from single view images/videos.
>
> (2) Real3D-Portrait: primarily designed for talking face generation.
>
> While effective for their tasks, these methods are 3d generative model --EG3D [6]variants, and tasks are momentary. They lack the ability to construct a personalized space for full-scale animation across life stages. We've compared with GANAvatar, a representative work that shares similar 3d generative model benefits (in Sec.5.2 Lines 1130-1144, Tab. 7 & Fig. 14),  while the results cannot preserve identity across ages or achieve consistent quality under the same training data conditions.
>
> We encourage the reviewer to show the evidence that proves these methods align more closely with our research goals or can achieve personalized, animatable head modeling as comprehensively as our approach with only using the same training data conditions.
>
> >**Q5**: How fast is the rendering of this method during inference?
>
> **A5**: Approximately 5-6 FPS on a single A100 GPU. This is improvable through further optimizations, e.g., adopting a C++ implementation of insta-ngp and leveraging libraries like tiny-cuda-nn to accelerate MLP networks. While rendering speed is not the primary focus of this work, we demonstrate superior performance and efficiency compared to recent reconstruction methods (see Sec. A.5.3, Lines 1215-1220).

---

> ### Author Response · Authors · 2024-11-28
> **Technical response - Part 2**
>
> >**Q6**: The method does not evaluate the identity preservation of the person when controlling for lifestage and shape.
>
> **A6**: We acknowledge that the current work does not explicitly evaluate identity preservation.  ArcFace [1] could be used  for obtaining quantitative evaluations. We will incorporate ID preservation assessment in future revisions. Additionally, we expect the reviewer can point out specific visual results where identity preservation appears to be compromised or worse than other methods, as this feedback will help refine our analysis.
>
> >**Q7**: When switching between different life stages, do you consider the order of switching time? Or multiple life stages switched randomly?
>
> **A7**: Both sequential and random switching between life stages can be supported by our method. After training, each life stage is associated with distinct learnable blending weights assigned to multiple multi-resolution hash grids. By adjusting these blending weight coefficients, we can achieve precise control over the progression of lifestages, ensuring controllability of the lifestage transitions. Alternatively, by setting random blending weights, the model can produce varied outcomes across different life stages.
>
>
> >**Q8**: It seems that the Gaussian and 3DMM of this method are bound together (during initialization). Why is it that when it is driven by moment (change the expression), it does not directly use 3DMM to deform the Gaussian points, but introduces a warp field instead?
>
> **A8**: The Gaussian Surfels are aligned with 3D morphable head model at initialization. However, in our personalized space construction, which accommodates varied appearances across life stages, strictly binding  during training would limit adaptability. To enable learning features like long hair or other life-stage-specific traits, we relax the Gaussian points from the 3D morphable head model. Fundamentally, our head avatar is still driven by FLAME expression/shape parameters. However, the absence of a direct mapping between Gaussian Surfels and the FLAME template make it unfeasible to directly employ morphable models for deforming the Gaussian points. To address this challenge, we introduce an inverse warping operation adapted from the methodology found in INSTA[4], in which details are unfolded on Sec.3.3.1 Lines 355-368.
>
> >**Q9**: In Table 5 (in 1 vs N), FlashAvatar reports lower results than INSTA. The experiments (1vsN) has a similar setting to FlashAvatar paper, but the results is inconsistent with the statement in the FlashAvatar paper that FlashAvatar performs better than INSTA. What could be the possible reason?
>
> **A9**: While the experimental setup for Table 5 (1 vs. N comparison) aligns with the FlashAvatar paper, the key difference lies in the dataset: for Table 5, we used Timewalker-1.0, which differs significantly from the datasets used in FlashAvatar.
>
> Timewalker-1.0 is composed of **unstructured data**, unlike the high-quality, front-view **"standard inputs"** used in FlashAvatar. Variations in data distribution, head poses, expressions, and noise levels likely explain the differences in results.
>
> To evaluate robustness and generalization, we conducted additional experiments with "standard inputs" data (Table 6). In this scenario, FlashAvatar outperformed INSTA, consistent with the findings reported in the FlashAvatar paper.
>
>
> **Citation**
> ```
> [1] Deng J, Guo J, Xue N, et al. Arcface: Additive angular margin loss for deep face recognition. In CVPR2019.
> [2] Zheng Y, Abrevaya V F, Bühler M C, et al. Im avatar: Implicit morphable head avatars from videos. In CVPR2022.
> [3] Zheng Y, Yifan W, Wetzstein G, et al. Pointavatar: Deformable point-based head avatars from videos. In CVPR2023.
> [4] Zielonka W, Bolkart T, Thies J. Instant volumetric head avatars. In CVPR2023.
> [5] Qian S, Kirschstein T, Schoneveld L, et al. Gaussianavatars: Photorealistic head avatars with rigged 3d gaussians. In CVPR2024.
> [6] Chan E R, Lin C Z, Chan M A, et al. Efficient geometry-aware 3d generative adversarial networks. In CVPR2022.
> ```

---

### Official Review · Reviewer_f3xJ · 2024-11-04

**Soundness:** 2
**Presentation:** 3
**Contribution:** 2
**Rating:** 6
**Confidence:** 5

**Summary:**

(Updated review; it ignores similarities with workshop paper)

The paper presents TimeWalker, a neural parametric representation that learns personalized representations of subjects with disentangled shape, expressions, and dynamic appearances across time.
The main contributions are twofold: a neural module called Dynamo to encode in a compact manner the main modes of variation in identity across time (a.k.a. a life stage), and another module called DNA-2DGS to encode expression deformations embedded into 3D Gaussians.

While the proposed method utilizes ideas from previous work, such as a canonical shape and appearance space, multi-resolution hash grids, and Gaussian surfels, the proposed modules have some merit and extend beyond previous methods. For instance, an adaptive multiresolution hash grid is proposed to encode neural head bases more efficiently. Also, the Gaussian surfels are adapted to a dynamic setup.

However, the proposed approach exhibits notorious artifacts in different facial regions, and it also shows lighting-identity entanglement due to a simple model that neglects lighting information from the underlying FLAME model. As such, the soundness of the approach is marginal. My final rating is borderline, somewhat inclined to acceptance, if an extended limitations section, additional comparisons, and an extended discussion with related work are added to the paper, and if the societal negative impacts and ethically sourced data are adequately addressed in the paper.

**Strengths:**

- A new module, called Dynamo, which utilizes a shared canonical shape space to encode identity and a deformed shape and appearance space to capture distinct colors and geometry from different videos captured at various times.
- A new module, called DNA-2DGS, which encodes dynamic expression deformation within a lifelong setting.
- The use of an adaptive multi-resolution hash grid to encode shape more efficiently with better generalization capabilities.
- The paper showcases a prompt-based 3D human head editing, demonstrating the Gaussian-based approach's generalization capabilities and flexibility.

**Weaknesses:**

- Lighting effects are not disentangled from life stages. In fact, it doesn’t leverage the lighting model of the underlying 3DMM-based model, i.e., FLAME. As a result, the lighting is part of the life stage, which is conceptually wrong.
Noticeable artifacts appear across facial feature boundaries, including eyelids and, more notoriously, the face outline. This could be likely ascribed to a lack of 3D Gaussian scaling regularization, especially across mesh boundaries in the UV space. A better dataset curation, e.g., using images only with good segmentation masks, could have handled this problem, at least for the face outline.
- Mouth interior reconstruction is poor, especially for teeth, which might be solved with more data and better modeling of the mouth interior, e.g., using a separate, more regularized appearance model. A better regularization of the surface normal, e.g., small normal deviations across neighboring surfel locations, could also alleviate the problem.
- Protocols 1 and 2 lack a more comprehensive comparison with other recent related work in 3D human head reconstruction at different ages and shapes. See the Questions section below for more details.
- The proposed dataset might raise ethical concerns as celebrities and actors have not consented to using their images and videos to generate and manipulate sensitive content, such as their faces. Discussions around ethically sourced data, e.g., the origin of the data, licensing, and consent, should be addressed and discussed.
- The paper must further address the potential societal negative impact, especially around Deepfakes. The authors should discuss additional ways to mitigate misuse (see Questions section for more details).

**Questions:**

*Section 3
- The introduction of the method in Section 3 (first two paragraphs) is redundant as most of it is already mentioned in the introduction. Please remove the first two paragraphs to release some space.
- Please move Section 3.1 (preliminaries) to the supplementary material, as more and more researchers are familiar with the basics of Gaussian Splatting.
The two suggestions above will give extra space to add more comparisons in the main paper.

*Comparisons
- Please move the comparisons with SoTA and mesh comparisons from the supplemental to the main paper.
- Additional comparison for protocols 1 and 2. Please add comparisons to the methods mentioned below.

*References and related work
- Please add the following references
1) Closely related work: Caliskan et al. 2024. PAV: Personalized Head Avatar from Unstructured Video Collection. ECCV 2024
The former paper is a dynamic NeRF-based approach that models an actor’s appearance and shape changes from an unstructured video collection. It utilizes a shared canonical shape space to encode identity and a deformed shape and appearance space to capture distinct colors and geometry from different videos captured at various times. Shape and appearance deformation are encoded through a multi-resolution hash grid and dynamic neural textures, respectively. The approach can naturally be adapted to model lifelong head avatars. Please highlight the differences between the approaches above and provide quantitative and qualitative comparisons, mainly for completeness, as this approach is closely related to the proposed method.

2) Approaches for comparison with Protocol 1 (at least one of the two below):
- Xu et al. 2024. Gaussian Head Avatar: Ultra High-fidelity Head Avatar via Dynamic Gaussians. CVPR 2024.
- Shao et al. 2024. SplattingAvatar: Realistic Real-Time Human Avatars with Mesh-Embedded Gaussian Splatting. CVPR 2024.
3) One-shot approaches for comparison with protocol 2 (at least one of the two below). These methods just require a reference image at different life stages to be able to create an avatar at a specific age.
- Ye et al. 2024. Real3D-Portrait: One-shot Realistic 3D Talking Portrait Synthesis. ICLR 2024.
- Chu et al. 2024. GPAvatar: Generalizable and Precise Head Avatar from Images. ICLR 2024.

*Societal negative impact
- Please include an extended section on ethical considerations about the dataset and target applications. This could cover ethically sourced data concerns (data origins, licensing, and consent) as well as potential misuse scenarios, e.g., deepfakes. For the latter, please discuss additional mitigation strategies or guidelines for responsible use of the technology and detection of DeepFakes. Please note that actors have not given consent to the use of their data. So, it is important to make sure the data is crawled with a Creative Commons license.

**Details Of Ethics Concerns:**

*Research transparency
- For full transparency, the authors should have attached the workshop paper to this submission. The above is good research practice, generally speaking.

*Societal negative impact/potentially harmful applications
- Potential societal negative impacts around ethically sourced data (data origins, licensing, and consent) and misuse (strategies and guidelines for mitigating and detecting DeepFakes) must be addressed further in this paper (see Questions section for specific concerns). Please note that actors haven’t given consent to the use of this data. So, ensuring that the videos are crawled with a Creative Commons license is a must.

---

> ### Author Response · Authors · 2024-11-28
> **Technical response**
>
> >**Q1**: Main contributions are two losses: lpips and consistency loss.
>
> **A1**: At no point in the paper we claim LPIPS or consistency loss as our contributions. These metrics are well-established, and commonly utilized strategies in the field, which are appropriately cited.
>
> >**Q2**: The application demonstrates the Gaussian-based approach's generalization capabilities and flexibility.
>
> **A2**: Interpreting this evaluation as a demonstration of GS approach’s flexibility misrepresents the paper’s clear purpose. The purpose of this evaluation is clear: to demonstrate the benefits of our neural parametric model, which achieves 3D-consistent and personalized outcomes. It is worth noting that DGE (Chen et al.) itself relies on GS as its base representation. The side-by-side comparison (Figure 17) illustrates the limitations of GS+DGE, such as incoherent geometry and blur, and how our method (Ours+DGE) addresses these issues.
> Above are all explicitly stated in Sec. A6.
>
> >**Q3**: Comparison with the recent work mentioned by reviewer- PAV: Personalized Head Avatar from Unstructured Video Collection.(ECCV2024)
>
> **A3**: (1) Note that PAV was released on 22nd, July, 2024,  making it a concurrent work with ours. (2) PAV builds on INSTA by extending it to support multiple appearances through the introduction of appearance embeddings. In our paper, we have already implemented and compared INSTA++, an enhanced version of INSTA, which integrates learnable latent factors into the MLP (Lines 1015–1017, Section A.5.2). This extension allows INSTA++ to model transitions across multiple life stages, aligning in spirit with PAV and enabling a fair and meaningful comparison. For a comprehensive analysis of the results, refer to Fig.11 & Tab.5 & Lines 1074-1129.
>
> >**Q4**: Societal negative impact/potentially harmful applications.
>
> **A4**: Considering the potential societal negative impacts around sourced data, we strictly obey the non-commercial license (Refer to Sec.A.7). While the real and rich human head assets generated by our method could be misused for deceptive applications, such as fake talking head video generation, they also have the potential to be leveraged for positive uses, such as training robust deepfake detection models. To mitigate risks, we will only make the dataset available to individuals or institutions who have reviewed and agreed to our proposed data usage agreement, ensuring ethical and responsible use.

---

### Author Response · Authors · 2024-11-28
**Declaration**

Declaration:

1. 20 IDs/40IDs: We explicitly state in the paper that the experimental data used is a subset of the TimeWalker dataset (e.g., line 475-478 outlines data usage for ablation studies, and Appendix (line 1058-1074) describes the evaluation protocol). Consequently, the experimental results are independent of whether the dataset comprises 20 IDs or 40 IDs.
-----------------

2. We strongly refute the baseless misinterpretations, accusations of plagiarism/dual submission, and instigative remarks made by some of the reviewers (**Reviewer f3xJ/TANo**). These allegations are entirely unfounded and constitute a direct attack on the integrity of our research, causing undue harm and insult to our research reputation.


3. Additionally, we also want to question the motivations and ethics behind actions of searching for the paper’s title externally. Such behavior constitutes a clear violation of the dual-blind review process and contradicts the fundamental principles of fairness and impartiality expected in academic peer review.


4. In adherence to the principles of scientific research, we will only address the technical inquiries as outlined below.

---

### Author Response · Authors · 2024-12-04
**Grave Misconduct of Duty by Reviewers in the Review Process**

Some of your (Reviewer f3xJ/TANo) handling of our submission has been deeply unprofessional and unacceptable. You initially twisted facts, made baseless accusations, issued malicious comments, and even incited other reviewers (Reviewer f3xJ to g9vf). These actions were not only damaging but also humiliating. **Despite the Area Chair’s clarification of the truth and explicit instructions to fairly re-evaluate the work, you have chosen to remain silent and disengaged during the re-review process**.

This deliberate inaction demonstrates a lack of respect for the scientific process and an unwillingness to address unjustified criticisms. Such behavior reflects clear bias and constitutes a blatant violation of your responsibilities as reviewers. It is deeply a discredit to the community to have reviewers behave in this manner.

We are unequivocal: this submission will not be withdrawn under any circumstances. Let the broader research community judge this situation and see how unwarranted drama and misconduct have undermined the ethical foundations of academic discourse.

---

### Note · Authors · 2025-01-22

**Comment:**

Unfair and unprofessional review process:

Some of the reviewers engaged in:

(1) repeated and deliberate misrepresentation of facts, malicious accusations (falsely claiming plagiarism and fabricating design flaws);

(2) outright refusal to re-review during the discussion period, despite the Area Chair’s clarification of the facts and explicit instructions to require the reviewers to re-evaluate our submission asap;

(3) remaining silent and disengaged during the whole discussion period.

(4) initiating a so-called "re-review" only after the discussion period ended—when we, as authors, could no longer respond;

(5) deliberate distortions and malicious comments:

(5.1) in the re-review phase, Reviewer f3xj completely ignored our replies and maliciously demanded comparisons with concurrent work PAV (which was just released on 22nd, July, 2024, and we already explained in the rebuttal)

(5.2) Reviewer f3xj  gave a fabricated critique alleged UV space artifacts, despite our framework containing no components involving UV space.

(5.3) Reviewer f3xj  suggested that FLAME’s lighting model could help solve in-the-wild illumination issues, reflecting a fundamental misunderstanding of FLAME’s limitations.

(5.4) Reviewer TANo reject our paper ("strong reject" in the inital stage, and "reject" in the re-review) with requiring us to compare with one-shot generative approaches without evidences, while our setting is reconstruction pipeline, not generative model, and we already selected a representative generative model to compare in the initial version for border and more comprehensive analysis.

(5.5) Reviewer f3xJ contradicting their own initial review by attacking the performance of our results, despite previously praising the “natural” outcomes of the same approach while falsely accusing us of plagiarizing our own abstract. This blatant inconsistency highlights the reviewer’s deliberate and malicious intent.

(6) Reviewer g9vf didn't update the re-review.

We acknowledge that as the baseline for this new task, there is space to improve the visual effects and lighting disentanglement, even our results are better than the compared methods. However, we didn't lack the comprehensive comparison with necessary SOTAs -- the remain ones the reviewers asked to add are either concurrent work without open source code ( while we already implemented an alternative version to compare), or focus on one-shot talking head task for face-reenactment on a specific appearance.

**Withdrawal Confirmation:**

I have read and agree with the venue's withdrawal policy on behalf of myself and my co-authors.